# A New Gain-Phase Error Pre-Calibration Method for Uniform Linear Arrays

**DOI:** 10.3390/s23052544

**Published:** 2023-02-24

**Authors:** Chang Liu, Xiao Tang, Zhi Zhang

**Affiliations:** School of Electrical Engineering & Intelligentization, Dongguan University of Technology, Dongguan 523808, China

**Keywords:** uniform linear arrays (ULAs), gain-phase error calibration, DOA estimation, weighted total least-squares (WTLS) problem, adaptive antenna nulling technique

## Abstract

In this paper, we consider the gain-phase error calibration problem for uniform linear arrays (ULAs). Based on the adaptive antenna nulling technique, a new gain-phase error pre-calibration method is proposed, requiring only one calibration source with known direction of arrival (DOA). In the proposed method, a ULA with *M* array elements is divided into M−1 sub-arrays, and the gain-phase error of each sub-array can be uniquely extracted one by one. Furthermore, in order to obtain the accurate gain-phase error in each sub-array, we formulate an errors-in-variables (EIV) model and present a weighted total least-squares (WTLS) algorithm by exploiting the structure of the received data on sub-arrays. In addition, the solution to the proposed WTLS algorithm is exactly analyzed in the statistical sense, and the spatial location of the calibration source is also discussed. Simulation results demonstrate the efficiency and feasibility of our proposed method in both large-scale and small-scale ULAs and the superiority to some state-of-the-art gain-phase error calibration approaches.

## 1. Introduction

Array signal processing has been intensively investigated over the last several decades [1,2]. As one of the hotspots in array signal processing, the direction of arrival (DOA) estimation problem of plane waves impinging on an antenna array has drawn widespread attention. Numerous DOA estimation algorithms have been developed to achieve high-resolution performance, e.g., multiple signal classification (MUSIC) [3], estimating signal parameters via rotational invariance technique (ESPRIT) [4], maximum likelihood (ML) approach [5], compressed sensing (CS) method [6] and sparse Bayesian learning (SBL) [7]. These estimators work well under the assumption that prior knowledge of the array manifold is available. However, in practical applications, this assumption is unrealistic since a series of array imperfections including gain-phase errors, mutual coupling and sensor location errors, may affect the ideal array manifold and degrade significantly the DOA estimation accuracy [8,9,10,11,12]. Therefore, it is highly desirable to estimate and calibrate array imperfections before DOA estimation. We focus the paper on estimating the gain-phase errors of uniform linear arrays (ULAs). Typically, the gain-phase errors are caused by the following factors: gradual changes of the behavior of the sensor itself and the internal circuits (due to thermal effects, aging of components, etc.), changes in location of the array elements induced by the vibrating wing of an aircraft or a hydrophone array behind a ship, and changes to the environment around the sensor array [13]. Herein, we consider these three main origins of gain-phase errors and assume that the gain-phase error caused by these changes is in the presence of a constant since all the above-mentioned changes are tiny and slow in a short time.

In recent decades, lots of effort has been made to address the gain-phase errors of antenna arrays [13,14,15,16,17,18,19,20,21,22,23,24,25]. Existing gain-phase error calibration methods can be roughly divided into two categories: self-calibration (without calibration sources) and pre-calibration (with calibration sources). In general, the first type of gain-phase error estimation method, which can jointly estimate the DOA parameters and gain-phase errors, does not require other calibration sources with known directions. Hence, it is more suitable for practice and has evoked much research interest for the past few years [13,14,15,16,17,18]. In [13,14], a self-calibration iterative method based on the eigenstructure is proposed to simultaneously estimate the DOAs of source signals and calibrate gain-phase errors. However, the drawback of this method is that it may suffer from suboptimal convergence. The algorithm proposed in [15] provides a consistent estimate of the gain-phase errors from the ML perspective, but it requires the covariance matrix of the ideal array output to be known. By using the diagonal lines of the covariance matrix, a class of the gain-phase error calibration algorithm is presented for linear equispaced arrays (LEA) in [16], but the perfect covariance matrix is unavailable when the number of snapshots taken to obtain the covariance matrix is small [17]. Z. Liu et al. proposed a novel sparse Bayesian array calibration (SBAC) method for compensating for all the typical array imperfections [18], whereas the online estimation of numerous unknown parameters makes the SBAC more complicated. Many proposals have been conducted on the gain-phase error calibration problem for nonlinear arrays [19,20,21]. In [19], A. Liu et al. presented an eigenstructure DOA algorithm in the presence of gain-phase errors, which is based on the dot product of the array output and its conjugate. Nevertheless, it requires that at least two signals are spatially far from each other. A Hadamard product-based method [20], which performs independently of the phase errors and without the requirement of two signals being spatially far separated from each other, achieves better estimation performance with respect to [19]. However, it suffers from high computational load. In [21], W. Xie et al. extended the amplitude-only measurements-based technique [19,20] into central symmetric arrays (CSAs) and proposed an algorithm for jointly estimating the DOAs of noncircular sources and gain-phase errors. In addition, for underwater sensing tasks, the random acoustic fluctuations in the medium, such as a highly dynamic ocean, may introduce phase errors in the output of the ULA. To address this issue, a novel L1-norm (absolute-error) maximum projection principal component analysis (PCA) method in [22] is proposed to resist the uncertainty of random acoustic fluctuations. Dubrovinskaya et al. designed acoustic arrays of an arbitrary shape and a practical algorithm to achieve accurate DOA estimates under some array imperfections, such as the spatial ambiguity that must be compensated for [23].

Unlike the self-calibration method, the second type of method, named the pre-calibration method, requires calibration sources with known directions. The eigenstructure-based pre-calibration method with two calibration sources with known directions was developed in [24], which depends on the covariance differencing technique and iterative method. However, since the gain-phase errors are estimated by using the phase of each entry in the steering vector, the possible phase ambiguities need to be checked. Based on the elegant analysis of the Cramer–Rao bounds on calibration and source location accuracies under three different sensor location situations, the authors of [26] indicate that two calibration sources are needed to calibrate the planar array, and the gain-phase estimation errors tend to decrease as the calibrating signal strength or power increases. By utilizing a set of calibration sources in known locations, a ML calibration algorithm is presented in [27] and the array perturbation parameters can be solved by means of the least square method. Nonetheless, the unique determination (identifiability) of the perturbation parameters is not satisfied in this method. Based on the null characteristic of the MUSIC spectrum, the array sensor gain-phase error calibration problem can be formulated as a series of linear equations and solved by a constrained optimization [25]. When the number of the linear equation is larger than that of the gain-phase errors to be estimated, a unique solution can be obtained. However, plenty of calibration sources with known DOAs are needed to solve for the uncertainties of array gain-phase errors. Under the existence of array phase errors, a non-coherent DOA estimation method is proposed in [28,29] based on the modified version of the greedy sparse phase retrieval (GESPR) framework [30]. Nevertheless, multiple calibration sources are required to cope with the phase ambiguity problem, and the global minima cannot be guaranteed [31]. Compared with the self-calibration method, although the pre-calibration method with calibration sources is somewhat limited practically when numerous calibration sources are required, there are two reasons why we take it to be once again in the spotlight. First, the pre-calibration one, in general, is more computationally efficient than the self-calibration since it avoids the alternation between the DOA estimation and array calibration, and the uncertainty calibration procedures are accomplished one at a time. Second, the pre-calibration can provide more satisfactory calibrating accuracies than the self-calibration due to some prior knowledge of the calibration sources [27]. Therefore, to obtain both satisfactory calibrating accuracy and low-cost implementation, we intend to explore an accurate, low-cost, broadly applicable gain-phase error calibrating scheme, which can be available for both small-scale and large-scale ULAs.

The adaptive antenna nulling technique, also known as power-inversion adaptive array [31], was firstly proposed by Appelbaum [32]. Its primitive application was to reduce sidelobe levels in the unknown direction of interferences by weighting the received signal vectors to minimize the interference powers. Alternatively, it is only necessary to have the steering vectors in the directions of interest to be zeros, i.e., null steering. In [33], several constrained null steering algorithms based on adaptive antenna nulling array are introduced, including mainly constrained least-mean-square (CLMS) and QR-recursive least-squares (QR-RLS) algorithms, and some convergence behaviors of the corresponding algorithms are also provided. The infinite impulse response (IIR) adaptive nulling array structure is designed in [34], which consists of a number of shift-invariant subarrays and takes the outputs of previous subarrays as spatial feedback. Moreover, the null steering scheme has been applied in combating spatial acoustic feedback between the hearing aid loudspeaker and microphone(s) [35,36], in which the calculation of null steering beamformer coefficients can be formulated as a least-square (LS) optimization or a min-max optimization so as to be null in the direction of the acoustic feedback. As an extension of [36], the same authors present a soft constraint on designing the null steering beamformer coefficients [37] in order to cope with the incoming acoustic signal preservation and feedback cancellation simultaneously.

In this paper, unlike using nulls of the MUSIC spectrum, we present a new gain-phase error pre-calibration method for ULAs by exploiting the adaptive antenna nulling technique [33,34] and null steering algorithm [35,36,37]. Only one known-DOA calibration source is required. We divide a ULA with *M* array elements into M−1 sub-arrays, and the gain-phase errors of each sub-array can be uniquely extracted one by one. To be specific, for the *m*th sub-array, it is able to derive the gain-phase errors as the reciprocal of the cumulative product of a series of complex coefficients, which relate to the sub-array unperturbed null steering vector (SAUNSV). In order to reliably estimate the SAUNSV, ${\bar{w}}_{m}$, we formulate the received data structure of ULAs as an errors-in-variables (EIV) model, based on which a weighted total least-squares (WTLS) algorithm is presented. Furthermore, we show that (1) the optimal location of the unique calibration source is the normal direction of the ULA. (2) When the calibration signal-to-source signal power ratio (CSR) increases, the gain-phase error estimation performance improves. As a consequence, it requires the calibration signal power to be much larger than the source signal. Our main contributions are therefore as follows: (1) By using the adaptive antenna nulling technique, a new gain-phase error pre-calibration method with only one calibration source is presented. (2) We formulate an errors-in-variables (EIV) model and propose a WTLS algorithm in order to estimate the SAUNSV, which is a crucial factor for the gain-phase error estimation. (3) Some comparative statistical analyses on the solution to the WTLS problem are derived.

The paper is organized as follows. In Section 2, the signal model for gain-phase error estimation is established and the adaptive antenna nulling technique is reviewed. The proposed method is derived in Section 3. In Section 4, we formulate an EIV model, propose a WTLS algorithm for estimating the SAUNSV and give the statistical analyses on the solution to the WTLS. Section 5 describes the simulation results, and the conclusion is given in Section 6.

*Notation:* Superscripts ${\left(\cdot \right)}^{H}$, ${\left(\cdot \right)}^{T}$ and ${\left(\cdot \right)}^{*}$ stand for conjugate transpose, transpose and complex conjugate, respectively. Matrices and vectors are represented by bold upper-case and bold lower-case characters, respectively. The notation diag(a) means forming a diagonal matrix by using the vector a as its main diagonal entries, while diag(Λ) denotes the vector formed by the diagonal of Λ if Λ is a diagonal matrix. The notation E(·) denotes the mathematical expectation, tr(·) stands for the trace operator, ||·|| and |·| represent the Euclidean norm and absolute value, respectively. ⊗ denotes the Kronecker product. $I_{M}$ denotes a M×M identity matrix, $0_{M}$ is a M×1 column vector with all zero entries and $0_{MM}$ is a M×M matrix with all zero entries. vec(·) denotes the vectorization operator, which stacks all columns of a matrix one below the other to form a column vector.

## 2. Problem Formulation

### 2.1. Signal Model

We consider that *L* narrowband far-field uncorrelated signals from directions $\theta =[\theta_{0},\theta_{1},\cdots ,\theta_{L-1}{]}^{T}$ impinge on a ULA, which consists of *M* (L<M) isotropic antenna elements. The distance between two adjacent elements is d=λλ22, i.e., a half-wavelength, where λ is the wavelength of the source signal. Without loss of generality, we take the first array element as reference, and hence, the gain-phase errors of the ULA can be modeled as a diagonal matrix, i.e.,
(1)$$\begin{array}{cc} \Gamma & =\mathrm{diag}(\left[\begin{array}{cccc} \Gamma_{0}, & \Gamma_{1}, & \cdots & \Gamma_{M-1} \end{array}\right])\\ \; & =\mathrm{diag}(\left[\begin{array}{cccc} 1, & g_{1}e^{j\varphi_{1}}, & \cdots & g_{M-1}e^{j\varphi_{M-1}} \end{array}\right]), \end{array}$$
where $\Gamma_{0}=1$ and $\Gamma_{m}=g_{m}e^{j\varphi_{m}}$, $g_{m}$ and $\varphi_{m}$ are the corresponding gain and phase errors of the (m+1)th (m=1,2,⋯,M−1) array elements, respectively.

In the *t*th snapshot, the received signal vector is written as
(2)$$\begin{array}{cc} e(t) & =\sum_{l=0}^{L-1}\Gamma a\left(\theta_{l}\right)s_{l}\left(t\right)+n\left(t\right)\\ \; & =\Gamma A(\theta )s(t)+n(t), \end{array}$$
where $A\left(\theta \right)=[a\left(\theta_{0}\right),a\left(\theta_{1}\right),\cdots ,a\left(\theta_{L-1}\right)]$ is the array manifold corresponding to the source signals. $a\left(\theta_{l}\right)=[1,e^{j\pi \sin\theta_{l}},e^{j2\pi \sin\theta_{l}},\cdots ,e^{j\left(M-1\right)\pi \sin\theta_{l}}{]}^{T}$ denotes the steering vector attached to the *l*th (l=0,1,⋯.L−1) uncorrelated source signal, $s_{l}\left(t\right)$. $s\left(t\right)=[s_{0}\left(t\right),s_{1}\left(t\right),\cdots ,s_{L-1}{\left(t\right)]}^{T}$ with equal power $\sigma_{s}^{2}$ which can be defined as $E[|s_{l}\left(t\right)\left|\right]=\sigma_{s}^{2}$, and $n\left(t\right)=[n_{0}\left(t\right),n_{1}\left(t\right),\cdots ,n_{M-1}{\left(t\right)]}^{T}$ is the additive noise vector, which is assumed to be spatially and temporally white with zero mean and covariance matrix $E\left[n\left(t\right)n{\left(t\right)}^{H}\right]=\sigma_{n}^{2}I_{M}$, where $\sigma_{n}^{2}$ is the noise variance of each array element. In addition, the noise herein is assumed to be an internal noise model, which means that the noise level in each array element is not affected by the array responses [38,39]. If the external noise model is considered [16], the noise level changes with both the array responses and received signal level.

We herein utilize a unique calibration signal $r_{0}\left(t\right)$ with known DOA, $\gamma_{0}$, to estimate and calibrate gain and phase errors. The calibration signal can be turned on and off at our command, and the following assumption associated with the calibration signal holds.

**Assumption** **1.***The calibration signal*$r_{0}\left(t\right)$*is uncorrelated with other source signals and measurement noises,*$\gamma_{0}\neq \theta_{l}$.

In this case, the received signal vector of the ULA is expressed as
(3)$$\begin{array}{cc} x(t) & =\Gamma b\left(\gamma_{0}\right)r_{0}\left(t\right)+e\left(t\right)\\ \; & =\Gamma b\left(\gamma_{0}\right)r_{0}\left(t\right)+\Gamma A\left(\theta \right)s\left(t\right)+n\left(t\right), \end{array}$$
where $b\left(\gamma_{0}\right)=[1,e^{j\pi \sin\gamma_{0}},e^{j2\pi \sin\gamma_{0}},\cdots ,e^{j\left(M-1\right)\pi \sin\gamma_{0}}{]}^{T}$ represents the steering vector associated with the calibration source signal. It should be noted that there is a special case when $s\left(t\right)=0_{L}$, i.e., no source signals are received, $x\left(t\right)=\Gamma b\left(\gamma_{0}\right)r_{0}\left(t\right)+n\left(t\right)$, e(t)=n(t), and $0_{L}$ is a column vector with *L* zero entries.

### 2.2. Adaptive Antenna Nulling Technique

As shown in Figure 1, the adaptive antenna nulling array [33] is composed of *M* array elements, which are uniformly spaced. The received signals in the adaptive nulling array are weighted by a set of complex weights, which aim to form nulls in certain directions, e.g., direction of strong interferences and unwanted signals. Moreover, the complex weights can be adjusted by various adaptive updating rules [40,41].

We consider the unperturbed null steering vector $\omega =[1,\omega_{1},\cdots ,\omega_{M-1}{]}^{T}$ corresponding to the unique calibration source, and the array pattern is expressed as $p\left(\gamma \right)=\omega^{H}\Gamma b\left(\gamma \right)$. Based on the adaptive antenna nulling technique, the unperturbed null steering vector can be applied to be null in the calibration signal direction, $\gamma_{0}$, i.e.,
(4)$$p\left(\gamma_{0}\right)=\omega^{H}\Gamma b\left(\gamma_{0}\right)=\omega^{H}b^{\prime }\left(\gamma_{0}\right)=0,$$
where $\gamma_{0}\in \left(-90^{\circ },90^{\circ }\right)$ and $b^{\prime }\left(\gamma_{0}\right)=\Gamma b\left(\gamma_{0}\right)$ is defined as the perturbed steering vector with gain-phase errors. Note that the first entry of the null steering vector, ω, is constrained to 1 since we take the first array element as reference.

## 3. Proposed Method

We divide a *M*-element ULA into M−1 sub-arrays as shown in Figure 2. Each sub-array consists of two adjacent sensor elements, the first one of which can be reckoned as the reference. For the *m*th (m=1,2,⋯,M−1) sub-array, the unperturbed null steering vector and unperturbed steering vector associated with the calibration signal can be defined as $w_{m}={\left[\begin{array}{cc} w_{m-1}, & w_{m} \end{array}\right]}^{T}$ and $b_{m}\left(\gamma_{0}\right)={\left[\begin{array}{cc} e^{j\left(m-1\right)\pi \sin\gamma_{0}}, & e^{jm\pi \sin\gamma_{0}} \end{array}\right]}^{T}$, respectively. Consequently, the nulling array pattern for the *m*th sub-array can be expressed as
(5)$$p_{m}\left(\gamma_{0}\right)=w_{m}^{H}\Gamma_{m}b_{m}\left(\gamma_{0}\right)=0,$$
where $\Gamma_{m}=\mathrm{diag}\left(\left[\begin{array}{cc} \Gamma_{m-1}, & \Gamma_{m} \end{array}\right]\right)$.

The unperturbed null steering vector in (5) can be rewritten as $w_{m}=w_{m-1}{\bar{w}}_{m}=w_{m-1}{\left[\begin{array}{cc} 1, & {\bar{w}}_{m} \end{array}\right]}^{T}$, where ${\bar{w}}_{m}={\left[\begin{array}{cc} 1, & {\bar{w}}_{m} \end{array}\right]}^{T}$ is called the sub-array unperturbed null steering vector (SAUNSV) and w¯m=wmwmwm+1wm+1, we also have $w_{0}={\bar{w}}_{0}=1$ and $w_{1}={\bar{w}}_{1}$, which can be defined as the first reference entry of the SAUNSV. Thus, for the first sub-array (m=1), $b_{1}\left(\gamma_{0}\right)={\left[\begin{array}{cc} 1, & e^{j\pi \sin\gamma_{0}} \end{array}\right]}^{T}$, $\Gamma_{0}=1$ and $\Gamma_{1}$ can be derived from (5) by
(6)$$\Gamma_{1}=-\frac{e^{-j\pi \sin\gamma_{0}}}{w_{1}^{*}}=-\frac{e^{-j\pi \sin\gamma_{0}}}{{\bar{w}}_{1}^{*}}.$$

For the *m*th (m≠1) sub-array, the gain-phase error of the *m*th array element can be derived from (5) as
(7)$$\Gamma_{m}=-\frac{\Gamma_{m-1}w_{m-1}^{*}e^{-j\pi \sin\gamma_{0}}}{w_{m}^{*}}=-\frac{\Gamma_{m-1}e^{-j\pi \sin\gamma_{0}}}{{\bar{w}}_{m}^{*}}.$$

Combining (6) with (7), it yields
(8)$$\Gamma_{m}={\left(-1\right)}^{m}\frac{e^{-jm\pi \sin\gamma_{0}}}{\prod_{i=0}^{i=m}{\bar{w}}_{i}^{*}}.$$

Since (8) implies that the gain-phase errors are determined on a series of complex coefficients ${\bar{w}}_{0},{\bar{w}}_{1,}\cdots ,{\bar{w}}_{M-1}$ and the known DOA of the calibration source, $\gamma_{0}$, it is a vital task to obtain the SAUNSV, ${\bar{w}}_{m}={\left[\begin{array}{cc} 1, & {\bar{w}}_{m} \end{array}\right]}^{T}$, by using the received signals. We propose a WTLS algorithm to estimate them in Section 4. Then, we outline the proposed gain-phase error estimation scheme in Algorithm 1, in which the DOA estimation problem is also considered.
**Algorithm 1** Proposed gain-phase error estimation method1:Turn on the calibration signal, and generate one calibration signal $r_{0}\left(t\right)$ with known direction, $\gamma_{0}$.2:Collect *T* (T>N) snapshots of receiving signals on the ULA, x(0),x(1),⋯,x(T−1).3:Set m=1, initialize with $\Gamma_{0}=1$ and ${\hat{\bar{w}}}_{0}=1$.4:**for**m=1 to M−1 **do**5:   Estimate the second entry of the SAUNSV, ${\hat{\bar{w}}}_{m}$, by using the WTLS algorithm presented in Section 4.6:   Calculate the estimate of the gain-phase error for the *m*th array element, ${\hat{\Gamma }}_{m}$, by using ${\hat{\bar{w}}}_{0},{\hat{\bar{w}}}_{1,}\cdots ,{\hat{\bar{w}}}_{m}$ according to (8).7:**end for**8:Turn off the calibration signal ($r_{0}\left(t\right)=0$), collect other $T_{0}$ signal snapshots and calibrate the gain-phase errors depending on the estimating results in 4–7, and carry out the DOA estimation by using the MUSIC or other DOA estimation methods.

## 4. Estimation of the SAUNSV

As can be seen in (3), the received calibration source signal vector $\Gamma b\left(\gamma_{0}\right)r_{0}\left(t\right)$ is perturbed by the source signal plus noise vector e(t)=ΓA(θ)s(t)+n(t). Our primary objective in this section is to formulate an EIV model for each sub-array and propose a WTLS algorithm to estimate ${\bar{w}}_{m}$ and $\Gamma_{m}$.

### 4.1. EIV Model for the SAUNSV Estimation

For the *m*th (m=1,2,⋯,M−1) sub-array, the received signal sub-vector, $x_{m}\left(t\right)$, with size of 2×1 is written by
(9)$$\begin{array}{cc} x_{m}\left(t\right) & =x_{cm}\left(t\right)+e_{m}\left(t\right)\\ \; & =\Gamma_{m}b_{m}\left(\gamma_{0}\right)r_{0}\left(t\right)+e_{m}\left(t\right)\\ \; & =\Gamma_{m}b_{m}\left(\gamma_{0}\right)r_{0}\left(t\right)+\Gamma_{m}A_{m}\left(\theta \right)s\left(t\right)+n_{m}\left(t\right), \end{array}$$
where $x_{cm}\left(t\right)=\Gamma_{m}b_{m}\left(\gamma_{0}\right)r_{0}\left(t\right)$, $e_{m}\left(t\right)=\Gamma_{m}A_{m}\left(\theta \right)s\left(t\right)+n_{m}\left(t\right)$, $n_{m}\left(t\right)={\left[n_{m-1}\left(t\right),n_{m}\left(t\right)\right]}^{T}$, $A_{m}\left(\theta \right)={\left[\begin{array}{cc} a_{m-1}^{T}, & a_{m}^{T} \end{array}\right]}^{T}$ and $a_{m}=\left[\begin{array}{cccc} e^{jm\pi \sin\theta_{0}}, & e^{jm\pi \sin\theta_{1}}, & \cdots & e^{jm\pi \sin\theta_{L-1}} \end{array}\right]$ is the *m*th row of A(θ).

Suppose, without loss of generality, $s\left(t\right)\neq 0_{L}$. By collecting *N* snapshots of $x_{m}\left(t\right)$, three corresponding matrices can be formed as
(10)$$X_{0}\left(t\right)=\left[\begin{array}{cccc} x_{m}\left(t\right), & x_{m}\left(t+1\right), & \cdots & x_{m}\left(t+N-1\right) \end{array}\right],$$
(11)$$X_{c}\left(t\right)=\left[\begin{array}{cccc} x_{cm}\left(t\right), & x_{cm}\left(t+1\right), & \cdots & x_{cm}\left(t+N-1\right) \end{array}\right],$$
(12)$$E_{x}\left(t\right)=-\left[\begin{array}{cccc} e_{m}\left(t\right), & e_{m}\left(t+1\right), & \cdots & e_{m}\left(t+N-1\right) \end{array}\right].$$

In what follows, we omit the snapshot index *t* for clarity. According to the nulling array pattern (5) and received signal model (9), we obtain
(13)$$\begin{array}{cc} \; & w_{m}^{H}X_{c}=w_{m}^{H}\left[X_{0}+E_{x}\right]=w_{m}^{H}\Gamma_{m}b_{m}\left(\gamma_{0}\right)r_{0}\\ \; & ={\bar{w}}_{m}^{H}\Gamma_{m}b_{m}\left(\gamma_{0}\right)r_{0}=0_{N}^{T}, \end{array}$$
where $r_{0}={\left[\begin{array}{cccc} r_{0}\left(t\right), & r_{0}\left(t+1\right), & \cdots & r_{0}\left(t+N-1\right) \end{array}\right]}^{T}$.

Note that the first entry of ${\bar{w}}_{m}$ is unit, (13) can be rewritten as
(14)$$\begin{array}{cc} \; & {\left[X_{0}+E_{x}\right]}^{H}{\bar{w}}_{m}={\left[X_{0}+E_{x}\right]}^{H}\left[\begin{array}{c} 1\\ {\bar{w}}_{m} \end{array}\right]\\ \; & =\left\{\left[\begin{array}{cc} -y & x_{0} \end{array}\right]+\left[\begin{array}{cc} -e_{y} & e_{x} \end{array}\right]\right\}\left[\begin{array}{c} 1\\ {\bar{w}}_{m} \end{array}\right]=0_{N}, \end{array}$$
where −y and $x_{0}$ are two N×1 column vectors which represent the first and second columns of $X_{0}^{H}$, respectively. In the same way, $-e_{y}$ and $e_{x}$ are the first and second columns of $E_{x}^{H}$, respectively.

Thus, we have
(15)$$\left[x_{0}+e_{x}\right]{\bar{w}}_{m}=y+e_{y}.$$

It is obvious that (15) is a standard EIV model for the SAUNSV estimation, and stems from a series of studies by G. H. Golub [42] and S. Van Huffel [43]. The calibration source signal vectors $x_{0}$ and y are perturbed by the random error vectors $-e_{x}$ and $-e_{y}$, respectively, which have to be corrected by the corresponding terms in (15). Furthermore, according to the unconditional-model assumption (UMA) that the source signal s(t) is random [44], we give the following two assumptions.

**Assumption** **2.***The random vector*$e_{0}={\left[\begin{array}{cc} e_{x}^{T} & e_{y}^{T} \end{array}\right]}^{T}$*is assumed to be a stationary one with zero mean, and its covariance matrix is expressed by*(16)$$Q_{e}=\left[\begin{array}{cc} Q_{xx} & Q_{xy}\\ Q_{yx} & Q_{yy} \end{array}\right],$$*where*$Q_{e}$*is the covariance matrix of*$e_{0}$, $Q_{xx}$*and*$Q_{yy}$*represent the auto-covariance matrices of*$e_{x}$*and*$e_{y}$*respectively, and*$Q_{xy}$*denotes the cross-covariance matrix of*$e_{x}$*and*$e_{y}$.

**Assumption** **3.***The matrix*$E_{x}$, *which denotes the source signal plus noise, is assumed to be uncorrelated along the columns. This means the vectors*$e_{m}\left(t+i\right)$*and*$e_{m}\left(t+j\right)$ (i≠j; i,j∈{0,1,2,⋯,N−1}) *for different snapshots are temporally uncorrelated. Only the row-correlation in*
$E_{x}$
*is considered. Thus, the covariance matrix*
$Q_{e}$
*can be approximated by*
(17)$$Q_{e}=\left[\begin{array}{cc} \sigma_{xx}^{2}I_{N} & \sigma_{xy}^{2}I_{N}\\ \sigma_{xy}^{2}I_{N} & \sigma_{yy}^{2}I_{N} \end{array}\right],$$
*where*
$\sigma_{xx}^{2}$
*denotes the variance of the source signal plus noise in the mth array element, i.e.,*
$e_{x}$, $\sigma_{yy}^{2}$
*is the variance related to the vector*
$e_{y}$, *and*
$\sigma_{xy}^{2}$
*is the cross covariance between*
$e_{x}$
*and*
$e_{y}$.

### 4.2. Proposed WTLS Algorithm

As mentioned in the previous subsection, the random vectors $e_{x}$ and $e_{y}$ are unknown in the EIV model (15), so the problem statement is, given the observation vectors $x_{0}$ and y, simultaneously estimate the SAUNSV, ${\bar{w}}_{m}$, vectors $e_{x}$ and $e_{y}$. Then, we formulate it as the following constrained WTLS optimization problem: (18)$$\begin{array}{c} \left\{{\hat{\bar{}}w}_{m},{\hat{e}}_{x},{\hat{e}}_{y}\right\}:=\arg\min_{\bar{w},e_{x},e_{y}}e_{0}^{H}Q_{e}^{-1}e_{0},\\ s.t.\left[x_{0}+e_{x}\right]{\bar{w}}_{m}=y+e_{y}, \end{array}$$
where the weighted matrix $Q_{e}^{-1}$ is the inverse matrix of the covariance matrix $Q_{e}$.

Aiming to solve the problem (18), it is necessary to recast (18) as described in the following lemma.

**Lemma** **1.***The constrained WTLS optimization problem in (18) is equivalent to the unconstrained optimization problem*(19)$$\left\{{\hat{\bar{w}}}_{m},{\hat{e}}_{x},{\hat{e}}_{y}\right\}:=\arg\min_{\bar{w},e_{x},e_{y}}J_{m}={\left(y-{\bar{w}}_{m}x_{0}\right)}^{H}{\left(B_{0}Q_{e}B_{0}^{H}\right)}^{-1}\left(y-{\bar{w}}_{m}x_{0}\right),$$*where*$B_{0}=\left[\begin{array}{cc} {\bar{w}}_{m}⊗I_{N} & -I_{N} \end{array}\right]$*is a*N×2N*matrix with full row rank*.

**Proof.** See Appendix A.    □

To attain the stationary point of cost function $J_{m}$, we take its partial derivative with respect to ${\bar{w}}_{m}^{*}$ and set it to 0, i.e.,
(20)$$\begin{array}{c} \frac{\partial J_{m}}{\partial {\bar{w}}_{m}^{*}}=\frac{\partial {\left(y-x_{0}{\bar{w}}_{m}\right)}^{H}}{\partial {\bar{w}}_{m}^{*}}{\left(B_{0}Q_{e}B_{0}^{H}\right)}^{-1}\left(y-x_{0}{\bar{w}}_{m}\right)\\ +{\left(y-x_{0}{\bar{w}}_{m}\right)}^{H}\frac{\partial {\left(B_{0}Q_{e}B_{0}^{H}\right)}^{-1}}{\partial {\bar{w}}_{m}^{*}}\left(y-x_{0}{\bar{w}}_{m}\right)\\ +{\left(y-x_{0}{\bar{w}}_{m}\right)}^{H}{\left(B_{0}Q_{e}B_{0}^{H}\right)}^{-1}\frac{\partial (y-x_{0}{\bar{w}}_{m})}{\partial {\bar{w}}_{m}^{*}}=0. \end{array}$$

The first partial derivative on the right hand of (20) can be expressed by
(21)$$\frac{\partial {\left(y-x_{0}{\bar{w}}_{m}\right)}^{H}}{\partial {\bar{w}}_{m}^{*}}=-x_{0}^{H}.$$

Due to the fact that $\frac{\partial z}{\partial z^{*}}=0$ for a complex variable *z*, the third partial derivative on the right hand of (20) can be given by
(22)$$\frac{\partial (y-x_{0}{\bar{w}}_{m})}{\partial {\bar{w}}_{m}^{*}}=0.$$

Based on the derivative property of a complex-valued inverse matrix, i.e., $\frac{\partial D^{-1}}{\partial w_{m}^{*}}=-D^{-1}\frac{\partial D}{\partial w_{m}^{*}}D^{-1}$[45], where D is a complex matrix, we have
(23)$$\frac{\partial {\left(B_{0}Q_{e}B_{0}^{H}\right)}^{-1}}{\partial {\bar{w}}_{m}^{*}}=-{\left(B_{0}Q_{e}B_{0}^{H}\right)}^{-1}\frac{\partial (B_{0}Q_{e}B_{0}^{H})}{\partial {\bar{w}}_{m}^{*}}{\left(B_{0}Q_{e}B_{0}^{H}\right)}^{-1}.$$

Then,
(24)$$\frac{\partial (B_{0}Q_{e}B_{0}^{H})}{\partial {\bar{w}}_{m}^{*}}=\frac{\partial (B_{0})}{\partial {\bar{w}}_{m}^{*}}Q_{e}B_{0}^{H}+B_{0}Q_{e}\frac{\partial (B_{0}^{H})}{\partial {\bar{w}}_{m}^{*}}.$$

Note that
(25)$$\frac{\partial (B_{0})}{\partial {\bar{w}}_{m}^{*}}=0_{N\times 2N},$$
and
(26)$$\begin{array}{c} \frac{\partial (B_{0}^{H})}{\partial {\bar{w}}_{m}^{*}}=\frac{\partial {\left[\begin{array}{cc} {\bar{w}}_{m}^{*}I_{N} & -I_{N} \end{array}\right]}^{T}}{\partial {\bar{w}}_{m}^{*}}\\ ={\left[\begin{array}{cc} I_{N} & -0_{N\times N} \end{array}\right]}^{T}. \end{array}$$

Thus, (24) can be rewritten as
(27)$$\frac{\partial (B_{0}Q_{e}B_{0}^{H})}{\partial {\bar{w}}_{m}^{*}}=B_{0}Q_{e}^{\left(1\right)},$$
where the 2N×N matrix $Q_{e}^{\left(1\right)}={\left[\begin{array}{cc} Q_{xx}^{T} & Q_{yx}^{T} \end{array}\right]}^{T}$ is the partitioned matrix constructed from the first column to *N*th column of $Q_{e}$.

Inserting (27) into (23), we obtain
(28)$$\frac{\partial {\left(B_{0}Q_{e}B_{0}^{H}\right)}^{-1}}{\partial {\bar{w}}_{m}^{*}}=-{\left(B_{0}Q_{e}B_{0}^{H}\right)}^{-1}B_{0}Q_{e}^{\left(1\right)}{\left(B_{0}Q_{e}B_{0}^{H}\right)}^{-1}.$$

Then, by combining (21), (22) and (28), (20) can be rewritten by
(29)$$\frac{\partial J_{m}}{\partial {\bar{w}}_{m}^{*}}=-{\left(x_{0}+c_{m}\right)}^{H}{\left(B_{0}Q_{e}B_{0}^{H}\right)}^{-1}\left(y-x_{0}{\bar{w}}_{m}\right)=0,$$
where the N×1 vector, $c_{m}$, is defined as
(30)$$c_{m}={\left(Q_{e}^{\left(1\right)}\right)}^{H}B_{0}^{H}{\left(B_{0}Q_{e}B_{0}^{H}\right)}^{-1}\left(y-x_{0}{\bar{w}}_{m}\right).$$

Actually, by considering the relation between $c_{m}$ and $e_{0}$ in (67), we should note that $c_{m}=e_{x}$ and
(31)$$e_{y}={\left(Q_{e}^{\left(2\right)}\right)}^{H}B_{0}^{H}{\left(B_{0}Q_{e}B_{0}^{H}\right)}^{-1}\left(y-x_{0}{\bar{w}}_{m}\right)$$
where the 2N×N matrix $Q_{e}^{\left(2\right)}$ is the partitioned matrix constructed from the (N+1)th column to (2N)th column of $Q_{e}$.

Therefore, based on (29), ${\bar{w}}_{m}$ can be estimated iteratively by
(32)$$\begin{array}{c} {\hat{\bar{w}}}_{m}={\left[{\left(x_{0}+c_{m}\right)}^{H}{\left(B_{0}Q_{e}B_{0}^{H}\right)}^{-1}x_{0}\right]}^{-1}{\left(x_{0}+c_{m}\right)}^{H}{\left(B_{0}Q_{e}B_{0}^{H}\right)}^{-1}y, \end{array}$$
where ${\hat{\bar{w}}}_{m}$ is the estimate of the second entry of the SAUNSV in the *m*th sub-array.

As can be seen in (32), the SAUNSV estimation requires the covariance matrix $Q_{e}$ to be known a priori, which is unrealistic. However, we can use the sample covariance matrix of the random vector $e_{0}$ instead of it. Hence, at the first stage of the proposed method, we must estimate the covariance matrix $Q_{e}$ when the calibration source signals are of no use, i.e., $r_{0}\left(t\right)=0$ in this stage. Specifically, for the *m*th (m=1,2,⋯,M−1) sub-array, we need to collect $K_{0}N$ ($K_{0}>1$) snapshots of receiving signals on the *m*th sub-array without the calibration source signals, $e_{m}\left(1\right),e_{m}\left(2\right),\cdots ,e_{m}\left(K_{0}N\right)$, and form $K_{0}$ random error vectors $e_{0}\left(0\right),e_{0}\left(1\right),\cdots ,e_{0}\left(K_{0}-1\right)$. Then, the covariance matrix $Q_{e}$ can be estimated by
(33)Q^e=(11K0)K0)∑k=0K0−1e0(k)e0H(k).

As a result, the steps of the proposed WTLS are tabulated as Algorithm 2.
**Algorithm 2** Proposed WTLS 1:Turn off the calibration signal ($r_{0}\left(t\right)=0$), collect $K_{0}N$ ($K_{0}>1$) snapshots of receiving signals, estimate the covariance matrices for M−1 sub-arrays, ${\hat{Q}}_{e,1},{\hat{Q}}_{e,2},\cdots ,{\hat{Q}}_{e,M-1}$ by using (33). 2:Turn on the calibration signal ($r_{0}\left(t\right)\neq 0$), set t=0, initialize with ${\hat{\bar{w}}}_{m}\left(0\right)=\alpha_{0}$, where $\alpha_{0}$ is a small positive parameter. 3:**repeat** 4:   Input the observation matrix $x_{0}\left(t\right)$ and vector y(t). 5:   Form the matrix $B_{0}\left(t\right)=\left[\begin{array}{cc} {\hat{\bar{w}}}_{m}{\left(t\right)}^{T}⊗I_{N} & -I_{N} \end{array}\right]$. 6:   Calculate the vector ${\hat{c}}_{m}\left(t\right)$ using (30). 7:   Estimate ${\hat{\bar{w}}}_{m}\left(t+1\right)$ using (32). 8:   set t=t+1. 9:**until**w¯^m(t+1)−w¯^m(t)w¯^m(t+1)−w¯^m(t)w¯^m(t)w¯^m(t)≤ε, ε is a given small positive threshold.10:Output the final SAUNSV estimate for the *m*th sub-array, ${\hat{\bar{w}}}_{m}={\left[\begin{array}{cc} 1, & {\hat{\bar{w}}}_{m}\left(t+1\right) \end{array}\right]}^{T}$.

The WTLS problem solver is derived from (19), which is only the necessary condition on optimality in WTLS problem. As a consequence, our proposed algorithm may only converge to a local minimum point. In order to attain the strict minimum point, we need to consider the sufficient condition on optimality and the Hessian matrix associated with the cost function $J_{m}$, which is given by
(34)$$H=\left[\begin{array}{cc} \frac{\partial^{2}J}{\partial {\bar{w}}_{m}^{*}\partial {\bar{w}}_{m}} & \frac{\partial^{2}J}{\partial^{2}{\bar{w}}_{m}^{*}}\\ \frac{\partial^{2}J}{\partial^{2}{\bar{w}}_{m}} & \frac{\partial^{2}J}{\partial \bar{w}\partial {\bar{w}}_{m}^{*}} \end{array}\right].$$

As is known, the sufficient condition of the desired minimum point is that the Hessian matrix H is positive definite. In practice, whether the Hessian matrix is positive definite is unknown since the TLS optimization is actually a non-convex problem [46,47,48,49]. However, fortunately, the LS solution can be commonly taken as the initial value of the iterative algorithm, and this is able to enhance the convergence efficiency. Furthermore, the numerical simulation results in this paper demonstrate empirically that the gain-phase error and DOA estimators work well, even when these local minima are introduced into the proposed gain-phase error estimation scheme.

### 4.3. Solution to the WTLS Problem

Let us consider the special case that $s\left(t\right)=0_{L}$, and the WTLS problem is degenerated into the conventional total least square (TLS) problem, i.e., the unconstrained objective function for the *m*th (m=1,2,⋯,M−1) sub-array in (19) can be expressed as
(35)$$\begin{array}{cc} J_{m} & ={\left(y-{\bar{w}}_{m}x_{0}\right)}^{H}{\left(B_{0}Q_{e}B_{0}^{H}\right)}^{-1}\left(y-{\bar{w}}_{m}x_{0}\right)\\ \; & =\frac{\sigma_{n}^{-2}{\left(y-{\bar{w}}_{m}x_{0}\right)}^{H}\left(y-{\bar{w}}_{m}x_{0}\right)}{1+{\left|{\bar{w}}_{m}\right|}^{2}}\\ \; & =\frac{\sigma_{n}^{-2}\left[\begin{array}{cc} 1, & {\bar{w}}_{m}^{*} \end{array}\right]X_{0}X_{0}^{H}{\left[\begin{array}{cc} 1, & {\bar{w}}_{m} \end{array}\right]}^{T}}{1+{\left|{\bar{w}}_{m}\right|}^{2}}, \end{array}$$
where $Q_{e}=\sigma_{n}^{2}I_{2N}$.

**Lemma** **2.***For the mth (*m=1,2,⋯,M−1*) sub-array and with no source signals (*$s\left(t\right)=0_{L}$*), the exact SAUNSV can be derived as*w¯m=[1,α1(2)α1(2)α1(1)α1(1)]T*, where*$\alpha_{1}\left(1\right)$*and*$\alpha_{1}\left(2\right)$*denote the first and second entries of the eigenvector with respect to the smallest eigenvalue of the received signal ensemble covariance matrix*$E[x_{m}x_{m}^{H}]$.

**Proof.** In the case of no source signals, the ensemble covariance matrix $E[x_{m}x_{m}^{H}]$ can be decomposed into
(36)$$\begin{array}{cc} E[x_{m}x_{m}^{H}] & =E\left[x_{cm}x_{cm}^{H}\right]+E\left[n_{m}n_{m}^{H}\right]\\ \; & ={\left[\begin{array}{cc} \alpha_{0} & \alpha_{1} \end{array}\right]}^{H}\left[\begin{array}{cc} \lambda_{0} & 0\\ 0 & \lambda_{1} \end{array}\right]\left[\begin{array}{cc} \alpha_{0} & \alpha_{1} \end{array}\right], \end{array}$$
where $\lambda_{0}=\sigma_{c}^{2}+\sigma_{n}^{2}$, $\lambda_{1}=\sigma_{n}^{2}$, $\sigma_{c}^{2}$ is the variance of the calibration signal, $\alpha_{0}$ and $\alpha_{1}$ represent the corresponding eigenvectors with respect to $\sigma_{c}^{2}+\sigma_{n}^{2}$ and $\sigma_{n}^{2}$, respectively.In essence, $\alpha_{1}$ is the noise sub-space, which consists of only one eigenvector, so we have $\alpha_{1}⊥\Gamma_{m}b_{m}\left(\gamma_{0}\right)$ and ${\bar{w}}_{m}\in \mathrm{span}\left(\alpha_{1}\right)$. Moreover, we note that the first entry of ${\bar{w}}_{m}$ is constrained to 1, thus ${\bar{w}}_{m}$ must be uniquely expressed as w¯m=[1,α1(2)α1(2)α1(1)α1(1)]T and w¯m=α1(2)α1(2)α1(1)α1(1).On the other hand, it is well known that the solution on the minimization of (35), i.e., the estimated SAUNSV, ${\hat{\bar{w}}}_{m}={\left[\begin{array}{cc} 1, & {\hat{\bar{w}}}_{m} \end{array}\right]}^{T}$, consistently converges to the exact solution provided that limN→∞X0X0HX0X0HNN exists and is positive definite [40]. Accordingly, when N→∞, we have X0X0HX0X0HNN→E[xmxmH] and it can be shown that for the estimate of (32), ${\hat{\bar{w}}}_{m}\to {\bar{w}}_{m}$. □

In the case that there exist source signals, i.e., $s\left(t\right)\neq 0_{L}$, the weighting covariance matrix $Q_{e}$ is not diagonal. We rewrite the objective function $J_{m}$ in (19) as
(37)$$\begin{array}{cc} J_{m} & ={\left(y-{\bar{w}}_{m}x_{0}\right)}^{H}{\left(B_{0}Q_{e}B_{0}^{H}\right)}^{-1}\left(y-{\bar{w}}_{m}x_{0}\right)\\ \; & =\frac{{\left(y-{\bar{w}}_{m}x_{0}\right)}^{H}\left(B_{0}B_{0}^{H}\right){\left(B_{0}Q_{e}B_{0}^{H}\right)}^{-1}\left(y-{\bar{w}}_{m}x_{0}\right)}{1+{\left|{\bar{w}}_{m}\right|}^{2}}\\ \; & =\frac{\left[\begin{array}{cc} 1, & {\bar{w}}_{m}^{*} \end{array}\right]X_{0}{\mathrm{PP}}^{H}X_{0}^{H}{\left[\begin{array}{cc} 1, & {\bar{w}}_{m} \end{array}\right]}^{T}}{1+{\left|{\bar{w}}_{m}\right|}^{2}}, \end{array}$$
where $B_{0}B_{0}^{H}=\left(1+\right|{\bar{w}}_{m}{|}^{2})I_{N}$, and the matrix P is defined as
(38)$$P=\sqrt{1+{\left|{\bar{w}}_{m}\right|}^{2}}{\left(B_{0}Q_{e}B_{0}^{H}\right)}^{-1/2}.$$

Substituting (17) into (38), we have (B0QeB0H)−1/2=[1[1(σxx2|w¯m|2+σyy2−2Re(w¯mσxy2))](σxx2|w¯m|2+σyy2−2Re(w¯mσxy2))]IN and ${\mathrm{PP}}^{H}$ can be expressed by
(39)PPH=(1+|w¯m|2)σxx2|w¯m|2+σyy2−2Re(w¯mσxy2)IN=κIN,
where the weighting factor κ=(1+|w¯m|2)[σxx2|w¯m|2+σyy2−2Re(w¯mσxy2)].

When N→∞, in the same way as (35), the solution on the minimization of (37), ${\hat{\bar{w}}}_{m}$, converges to a specific solution ${\bar{w}}_{m}^{\prime }$ asymptotically, where w¯m′=α1′(2)α1′(2)α1′(1)α1′(1), $\alpha_{1}^{\prime }\left(1\right)$ and $\alpha_{1}^{\prime }\left(2\right)$ denote the first and second entries of the eigenvector ${\alpha^{\prime }}_{1}={\left[\alpha_{1}^{\prime }\left(1\right)\begin{array}{cc} , & \alpha_{1}^{\prime }\left(2\right) \end{array}\right]}^{T}$ with respect to the smallest eigenvalue of the weighted sample covariance matrix, X0PPHX0HX0PPHX0HNN, which can be approximated as
(40)X0PPHX0HX0PPHX0HNN=κX0X0HκX0X0HNN→κE[xmxmH](N→∞)

Recalling Assumption 1, we can obtain
(41)$$E\left[x_{m}x_{m}^{H}\right]=E\left[x_{cm}x_{cm}^{H}\right]+E\left[x_{r}x_{r}^{H}\right]+E\left[n_{m}n_{m}^{H}\right]$$
where $x_{r}=\Gamma_{m}A_{m}\left(\theta \right)s\left(t\right)$.

From (39)–(41), we can observe that (1) the weighting matrix ${\mathrm{PP}}^{H}$ reduces to a diagonal matrix with the weighting factor κ, which enables the data from different columns of the matrix $E_{x}^{H}{\left[\begin{array}{cc} 1, & {\bar{w}}_{m} \end{array}\right]}^{T}$ to be equally sized. When $e_{x}$ and $e_{y}$ are equally sized and uncorrelated, i.e., $\sigma_{xx}^{2}=\sigma_{yy}^{2}$ and $\sigma_{xy}^{2}=0$, one obtains $\kappa =\sigma_{xx}^{-2}$ (For a special case that $s\left(t\right)=0_{L}$ and $E_{x}^{H}$ only depends on noises, $\kappa =\sigma_{xx}^{-2}=\sigma_{n}^{-2}$); (2) when N→∞, the estimated SAUNSV obtained by the WTLS problem (37) hinges on the eigen-decomposition of X0PPHX0HX0PPHX0HNN, which is tantamount to that of $\kappa E[x_{m}x_{m}^{H}]$. In the following, we shall conduct discussion on the specific solution, ${\bar{w}}_{m}^{\prime }$, in two cases of uncorrelated and correlated source signals, respectively.


**(1) Uncorrelated source signal case**


Let two Hermitian matrices be $Y=E\left[x_{cm}x_{cm}^{H}\right]+E\left[n_{m}n_{m}^{H}\right]$ and $Z=E[x_{r}x_{r}^{H}]$. The eigen-decomposition of Y leads to the eigenvector $\alpha_{1}$ corresponding to the exact SAUNSV in (36), and Z can be expressed as
(42)$$Z=E\left[x_{r}x_{r}^{H}\right]=\Gamma_{m}A_{m}\left(\theta \right)R_{s}A_{m}^{H}\left(\theta \right)\Gamma_{m}^{H}=\sigma_{s}^{2}Z_{0},$$
where $R_{s}=E\left[s\left(t\right)s^{H}\left(t\right)\right]=\sigma_{s}^{2}I_{L}$ and $Z_{0}=\Gamma_{m}A_{m}\left(\theta \right)A_{m}^{H}\left(\theta \right)\Gamma_{m}^{H}$. It is immediately obvious that the matrix Y is perturbed by $\sigma_{s}^{2}Z_{0}$, inevitably causing perturbations on $\alpha_{1}$.

Based on the matrix perturbation theory (see [50], pp. 69–70), the perturbed eigenvector ${\alpha^{\prime }}_{1}$ corresponding to $\alpha_{1}$ can be given by using a convergent power series
(43)$${\alpha^{\prime }}_{1}=\alpha_{1}+\left(\sum_{k=1}^{\infty }\mu_{k}\sigma_{s}^{2k}\right)\alpha_{0},$$
where the expression in bracket is a convergent power series provided that $\sigma_{s}^{2}$ is sufficiently small, and $\mu_{k}$ (k=1,2,⋯,∞) are the coefficients of the perturbation expansions of ${\alpha^{\prime }}_{1}$ along $\alpha_{0}$. The first- and second-order perturbation expansion coefficients are given as (see Appendix B)
(44)$$\mu_{1}=\frac{\alpha_{0}^{H}Z_{0}\alpha_{1}}{\left(\lambda_{1}-\lambda_{0}\right)\left|\right|\alpha_{0}{\left|\right|}^{2}}=-\frac{\alpha_{0}^{H}Z_{0}\alpha_{1}}{\sigma_{c}^{2}\left|\right|\alpha_{0}{\left|\right|}^{2}},$$
and
(45)$$\mu_{2}=\frac{\left(\alpha_{1}^{H}Z_{0}\alpha_{1}\right)\left(\alpha_{0}^{H}Z_{0}\alpha_{1}\right)}{\sigma_{c}^{4}\left|\right|\alpha_{0}{\left|\right|}^{2}\left|\right|\alpha_{1}{\left|\right|}^{2}}-\frac{\left(\alpha_{0}^{H}Z_{0}\alpha_{1}\right)\left(\alpha_{0}^{H}Z_{0}\alpha_{0}\right)}{\sigma_{c}^{4}\left|\right|\alpha_{0}{\left|\right|}^{4}}.$$

Subsequently, by keeping the first- and second-order terms of the source signal power $\sigma_{s}^{2}$, we obtain the perturbed term in (43) as
(46)$$\begin{array}{cc} \; & \Delta \alpha_{1}\approx \left(\mu_{1}\sigma_{s}^{2}+\mu_{2}\sigma_{s}^{4}\right)\alpha_{0}\\ \; & =-\frac{1}{\mathrm{CSR}}\frac{\alpha_{0}^{H}Z_{0}\alpha_{1}}{\left|\right|\alpha_{0}{\left|\right|}^{2}}\alpha_{0}+\frac{1}{{\mathrm{CSR}}^{2}}\frac{\left(\alpha_{1}^{H}Z_{0}\alpha_{1}\right)\left(\alpha_{0}^{H}Z_{0}\alpha_{1}\right)}{\left|\right|\alpha_{0}{\left|\right|}^{2}\left|\right|\alpha_{1}{\left|\right|}^{2}}\alpha_{0}\\ & -\frac{1}{{\mathrm{CSR}}^{2}}\frac{\left(\alpha_{0}^{H}Z_{0}\alpha_{1}\right)\left(\alpha_{0}^{H}Z_{0}\alpha_{0}\right)}{\left|\right|\alpha_{0}{\left|\right|}^{4}}\alpha_{0}, \end{array}$$
and the specific solution to the WTLS problem (37), ${\bar{w}}_{m}^{\prime }$, can be written by
(47)$${\bar{w}}_{m}^{\prime }=\frac{\alpha_{1}\left(2\right)+\Delta \alpha_{1}\left(2\right)}{\alpha_{1}\left(1\right)+\Delta \alpha_{1}\left(1\right)},$$
where CSR=σc2σc2σs2σs2 is called the calibration signal-to-source signal ratio, and $\Delta \alpha_{1}\left(1\right)$ and $\Delta \alpha_{1}\left(2\right)$ denote the first and second entries of $\Delta \alpha_{1}$, respectively.


**(2) Correlated source signal case**


In this case, Z can be partitioned as
(48)$$\begin{array}{cc} Z & =E\left[x_{r}x_{r}^{H}\right]=\Gamma_{m}A_{m}\left(\theta \right)R_{s}A_{m}^{H}\left(\theta \right)\Gamma_{m}^{H}\\ \; & =\sigma_{s}^{2}\Gamma_{m}A_{m}\left(\theta \right)A_{m}^{H}\left(\theta \right)\Gamma_{m}^{H}+\Gamma_{m}A_{m}\left(\theta \right){\mathrm{FA}}_{m}^{H}\left(\theta \right)\Gamma_{m}^{H}\\ \; & =\sigma_{s}^{2}{\bar{Z}}_{0}, \end{array}$$
where $R_{s}=\sigma_{s}^{2}I_{L}+F$, $\sigma_{s}^{2}I_{L}$ and F contain the diagonal and off-diagonal entries of matrix $R_{s}$, respectively. Matrix ${\bar{Z}}_{0}$ can be written as
(49)$${\bar{Z}}_{0}=Z_{0}+Z_{1},$$
where Z1=(11σs2σs2)ΓmAm(θ)FAmH(θ)ΓmH.

In the same way as the uncorrelated source case, we can derive the first-order perturbation expansion coefficient as
(50)$$\mu_{1}^{\prime }=\frac{\alpha_{0}^{H}{\bar{Z}}_{0}\alpha_{1}}{\left(\lambda_{1}-\lambda_{0}\right)\left|\right|\alpha_{0}{\left|\right|}^{2}}=-\frac{\alpha_{0}^{H}Z_{0}\alpha_{1}}{\sigma_{c}^{2}\left|\right|\alpha_{0}{\left|\right|}^{2}}-\frac{\alpha_{0}^{H}Z_{1}\alpha_{1}}{\sigma_{c}^{2}\left|\right|\alpha_{0}{\left|\right|}^{2}}.$$

From (50), we can note that the first-order perturbation expansion coefficient, compared to the uncorrelated source case in (44), is determined by both source signal auto-correlation and cross-correlation, which correspond to the first term and second term on the right hand of (50), respectively. Moreover, for the second-order perturbation expansion coefficient, we can find similar results by following (45). As a result, both the specific solution, ${\bar{w}}_{m}^{\prime }$, and the estimated gain-phase error in (8) are perturbed by the cross-correlation between the source signals.

### 4.4. Spatial Location of the Calibration Source

Since the SAUNSV derived from (37) is perturbed from its exact value, this, of course, causes a bias on the null position of the null-spectrum in (5). Let $\gamma_{0}^{\prime }=\gamma_{0}+\Delta \gamma_{0}$ be the practical null position when using ${\bar{w}}_{m}^{\prime }$, and $\Delta \gamma_{0}$ is the null position bias. Following (5), we have
(51)$$\begin{array}{cc} p_{m}\left(\gamma_{0}^{\prime }\right) & ={\bar{w}}_{m}^{{}^{\prime }H}\Gamma_{m}b_{m}\left(\gamma_{0}+\Delta \gamma_{0}\right)\\ \; & ={\bar{w}}_{m}^{{}^{\prime }H}\Gamma_{m}b_{1}\left(\gamma_{0}+\Delta \gamma_{0}\right)=0, \end{array}$$
where ${{\bar{w}}^{\prime }}_{m}={\left[\begin{array}{cc} 1, & {\bar{w}}_{m}^{\prime } \end{array}\right]}^{T}$, $b_{m}\left(\gamma_{0}+\Delta \gamma_{0}\right)={\left[\begin{array}{cc} e^{j\left(m-1\right)\pi \sin\left(\gamma_{0}+\Delta \gamma_{0}\right)}, & e^{jm\pi \sin(\gamma_{0}+\Delta \gamma_{0})} \end{array}\right]}^{T}$ and $b_{1}\left(\gamma_{0}+\Delta \gamma_{0}\right)={\left[\begin{array}{cc} 1, & e^{j\pi \sin(\gamma_{0}+\Delta \gamma_{0})} \end{array}\right]}^{T}$.

Let $\Delta {{\bar{w}}^{\prime }}_{m}={{\bar{w}}^{\prime }}_{m}-{\bar{w}}_{m}={\left[\begin{array}{cc} 0, & \Delta {\bar{w}}_{m}^{\prime } \end{array}\right]}^{T}$ be the difference between the vector ${{\bar{w}}^{\prime }}_{m}$ and ${\bar{w}}_{m}$. Moreover, it can be shown that
(52)$$\Delta {\bar{w}}_{m}^{\prime }=\frac{\alpha_{1}\left(1\right)\Delta \alpha_{1}\left(2\right)-\alpha_{1}\left(2\right)\Delta \alpha_{1}\left(1\right)}{\left[\alpha_{1}\left(1\right)+\Delta \alpha_{1}\left(1\right)\right]\alpha_{1}\left(1\right)}.$$

By expanding $b_{1}\left(\gamma_{0}+\Delta \gamma_{0}\right)$ in the first-order Taylor series around the ideal null position $\gamma_{0}$, we have
(53)$$b_{1}\left(\gamma_{0}+\Delta \gamma_{0}\right)\approx b_{1}\left(\gamma_{0}\right)+{\left[\begin{array}{cc} 0, & j\pi \cos\left(\gamma_{0}\right)e^{j\pi \sin(\gamma_{0})} \end{array}\right]}^{T}\Delta \gamma_{0}.$$

Substituting (53) and $\Delta {{\bar{w}}^{\prime }}_{m}$ into (51) yields
(54)$$\Delta \gamma_{0}=-{\left(\frac{\Delta {\bar{w}}_{m}^{\prime }}{{\bar{w}}_{m}^{\prime }}\right)}^{*}\frac{1}{j\pi \cos\gamma_{0}}.$$

It is of interest to note from (54) that in the case that the calibration source is located in the normal direction of the ULA ($\gamma_{0}=0^{\circ }$), the minimum null position bias, |Δγ0|=|Δw¯m′|Δw¯m′πw¯m′|πw¯m′| is obtained.

## 5. Simulation Results and Discussion

In this section, several simulations are conducted to verify the validity and effectiveness of the proposed gain-phase error estimation method. We first consider the gain-phase error estimation performance of the proposed method for both small-scale and large-scale ULAs in Section 5.1. Then, in Section 5.2, the proposed method is compared with the eigenstructure method [13], diagonal line method [17], and MUSIC nulling method [25]. At last, in Section 5.3, the DOA estimation performance results are provided, which depend on the aforementioned gain-phase error estimation methods. All the results are obtained by averaging over 500 Monte Carlo trials. Two far-field uncorrelated source signals, with equal power $\sigma_{s}^{2}$, impinge on the ULA from directions $5^{\circ }$ and $30^{\circ }$. The power of the unique calibration signal, $\sigma_{c}^{2}$ is set to be 1, which is located at $\gamma_{0}=0^{\circ }$. The signal-to-noise ratio (SNR) is defined as SNR(dB)=10log10σs2σs2σn2σn2. In addition, since one calibration signal coexists with the source signals in the gain-phase error estimation stage, we define the calibration signal-to-source signal ratio (CSR) as CSR(dB)=10log10σc2σc2σs2σs2.

The random gain-phase errors, which are proved to be feasible in [51], can be assumed to be Gaussian distribution, i.e., $g_{m}∼N\left(0,\sigma_{g}^{2}\right)$, where $\sigma_{g}^{2}$ is the variance of gain errors. In addition, for the phase error, we also have $\gamma_{m}∼N\left(0,\sigma_{\gamma }^{2}\right)$, where $\sigma_{\gamma }^{2}$ is the variance of phase errors. In simulations, $\sigma_{g}^{2}$ and $\sigma_{\gamma }^{2}$ are set to be 0.1 and 36, respectively. The deterministic gain-phase error is also considered in Section 5.2 and Section 5.3 for a comparison with the cited algorithms. In order to evaluate the estimation precision of the proposed method, the root-mean-square error (RMSE) of the gain-phase error estimation is used, which is defined as
(55)$${\mathrm{RMSE}}_{\Gamma }=\sqrt{\frac{1}{KM}\sum_{k=1}^{K}{∥\mathrm{diag}(\Gamma )-\mathrm{diag}({\hat{\Gamma }}^{\left(k\right)})∥}^{2}},$$
where *K* denotes the number of trials and ${\hat{\Gamma }}^{\left(k\right)}$ is the estimated gain-phase error in the *k*th trial.

Similarly, the RMSE for testing the DOA estimation performance is defined as shown below:(56)$${\mathrm{RMSE}}_{\theta }=\sqrt{\frac{1}{LK}\sum_{l=0}^{L-1}\sum_{k=1}^{K}{\left|\theta_{l}-{\hat{\theta }}_{l,k}\right|}^{2}},$$
where ${\hat{\theta }}_{l,k}$ is the estimated DOA of the *l*th source signal in the *k*th trial.

### 5.1. Gain-Phase Error Estimation Performance

In this subsection, we study the gain-phase error estimation performance results of the proposed method in terms of CSRs, snapshots, number of array elements, spatial location of the calibration signal, and correlation between the source signals. The simulated RMSEs are calculated depending on the proposed method, while the theoretical RMSE values are derived by using (47). Unless otherwise stated, the total number of snapshot T=80 and N=40. Accordingly, we can see that TTNN=2 iterations in WTLS are carried out.

Figure 3 shows the gain-phase error estimation RMSE for the proposed method versus CSRs. The numbers of array elements are set to M=6,8,12,16 for Figure 3a–d, respectively. Similar cases for large-scale ULAs, i.e., M=64,128,256,512 are illustrated in Figure 4. We can observe that the RMSEs decrease as CSRs increase, and more satisfactory gain-phase error estimation performance results can be achieved when the power of the calibration source is far larger than that of the signal source. This is due to statistical perturbation of the SAUNSVs according to (46), which depends on CSRs. Furthermore, it is evident that the simulation results closely agree with theoretical predictions when the CSR is larger than 20 dB.

In Figure 5, the RMSE of the gain-phase estimation using the proposed method is plotted versus numbers of snapshots *T* for different array elements M=1,050,100. Two iterations are carried out for the proposed method with N=T/2. It can clearly be seen that the RMSE decreases as *T* increases and increases by increasing *M*, which coincides with the explicit expression of the gain-phase error estimate (8). The more estimated coefficients ${\hat{\bar{w}}}_{m}$ involve, the larger the estimated error becomes. Consequently, the gain-phase error estimation accuracy and RMSE degrade with the increase in the number of array elements.

The gain-phase error estimation RMSE for different spatial locations of the calibration source, $\gamma_{0}$, is shown in Figure 6a. We observe that the RMSE performance of the proposed method degrades with the increase in $\gamma_{0}$, which attributes to the null position bias of the practical null spectrum in (51). Furthermore, the corresponding null position bias, $\Delta \gamma_{0}$, is plotted in Figure 6b, from which the simulation results are nearly close in agreement with the theoretical values calculated by (54). The minimum null position bias is achieved in the case of $\gamma_{0}=0^{\circ }$.

Figure 7 shows the gain-phase error estimation performance for different correlation factors of two source signals. In this case, the source covariance matrix can be expressed as
(57)$$R_{s}=E\left[s\left(t\right)s^{H}\left(t\right)\right]=\sigma_{s}^{2}\left[\begin{array}{cc} 1 & \xi \\ \xi^{*} & 1 \end{array}\right],$$
where ξ is the correlation factor. We can note that the RMSE slightly increases by increasing the correlation factor; however, this adverse impact can be reduced by raising the CSR.

### 5.2. Comparison to Other Calibration Methods

In this subsection, both deterministic and random gain-phase errors are considered. The deterministic gain-phase error is modeled as $\Gamma =\mathrm{diag}(\left[1e^{j0^{\circ }}2e^{j8.45^{\circ }}5e^{j12.75^{\circ }}3.2e^{-j15^{\circ }}0.2e^{j6.2^{\circ }}\right])$ with M=5, and the random gain-phase error is also subject to the Gaussian distribution with zero mean and variances given in Section 5.1. For a fair comparision, we also apply only one calibration signal located at $0^{\circ }$ for the MUSIC nulling method [25].

Figure 8a,b compare the gain-phase error estimation performance results of the corresponding methods for different SNRs in 5-element ULA with deterministic gain-phase errors and 16-element ULA with random gain-phase errors, respectively. The eigenstructure method [13] and diagonal line method [17] belong to the method of the self-calibration type, whereas the MUSIC nulling method [25] is a pre-calibration method like our proposed one. Generally, the pre-calibration methods are superior to the self-calibration type because of the application of calibration sources with known directions. The eigenstructure method [13] fails to work no matter how high the SNR is due to its suboptimal solutions and lack of unique properties. For the diagonal line method [17], an insufficient number of snapshots limits the estimation of the covariance matrix, resulting in a weak gain-phase error estimation performance. The proposed method behaves better than other competitive methods. The RMSE performance comparison on the corresponding methods with a different number of array element is shown in Figure 9. It is observed that the RMSEs of the MUSIC nulling and proposed methods increase as the number of array element increases because more parameters are needed to be estimated for larger arrays which causes larger estimating errors. Figure 10 presents the performance comparison for different numbers of snapshot *T* in 5-element ULA with deterministic gain-phase errors. For the eigenstructure method [13] and proposed method, two iterations are carried out with N=T/2. We can see that the proposed one performs better than others.

### 5.3. DOA Estimation Performance

In this subsection, we compare the DOA estimation performance results of the corresponding methods. The CRB on DOA estimation is also given, which is calculated by Equation (4.6) in ref. [52]. All the simulation scenarios are identical to Section 5.2. For estimating DOA of source signals, another $T_{0}=40$ signal snapshots are needed to be collected. For effective evaluation and comparison of the proposed method, the DOA estimation performance of the MUSIC algorithm in the absence of gain-phase errors is taken as the benchmark. When applying the eigenstructure method [13], the DOA can be jointly estimated with the gain-phase errors, while the MUSIC estimator is used to obtain the DOA results after the gain-phase errors are calibrated by the diagonal line method [17], MUSIC nulling method [25] and proposed method. Once the gain-phase error estimate $\hat{\Gamma }$ is available, the MUSIC spatial spectrum can be obtained by
(58)$$f\left(\theta \right)=\frac{1}{a^{H}\left(\theta \right){\hat{U}}_{e}{\hat{U}}_{e}^{H}a\left(\theta \right)},\theta \in \left[-90^{\circ },90^{\circ }\right]$$
where ${\hat{U}}_{e}$ is the gain-phase error-free noise subspace with dimension M×(M−L). It can be obtained after the gain-phase error calibration and extracted by using the eigenvalue decomposition (EVD) of the signal covariance matrix estimate, which is calculated by R^e=(1(1T0T0)∑t=T+1T+T0Γ^e(t)eH(t)Γ^H.

Finally, we can find the DOA estimate $\hat{\theta }$ by using the peak searching of f(θ). In order to reduce the grid mismatch, we herein make use of a recursive grid refinement searching scheme, which contains three steps: (1) Create the maximum and minimum angular grids, whose grid step sizes are $\theta_{s,\max}$ and $\theta_{s,\min}$, respectively, and obtain the rough DOA estimate by using $\theta_{s,\max}$. (2) Decrease the grid step size, θs,max=θs,maxθs,maxII and I>1, and obtain a refined DOA estimate by using the updated $\theta_{s,\max}$ at a local range around the rough DOA estimate in the last step. (3) Return to step (2) until $\theta_{s,\max}\leq \theta_{s,\min}$. In our simulations, $\theta_{s,\max}=1^{\circ }$, $\theta_{s,\min}=0.01^{\circ }$ and I=10. Figure 11a,b show the DOA estimation performance results of the proposed method and MUSIC algorithm versus CSR in 5-element ULA and 16-element ULA, respectively. It can be observed that the proposed method can achieve satisfactory DOA performance, like MUSIC, without gain-phase errors at high CSR.

Figure 12 and Figure 13 show the DOA estimation performance comparison among the corresponding methods in 5-element ULA and 16-element ULA, respectively. It is found that our proposed method offers similar DOA estimation performance results to MUSIC in the absence of gain-phase errors, owing to the fact that the gain-phase errors are correctly estimated and calibrated. Figure 14 provides the DOA estimation performance comparison for different numbers of the array element; we also can see the superiority of the proposed one. Figure 15 offers the DOA estimation performance comparison of different numbers of snapshots in the 5-element array. It can be noted that the RMSE decreases when increasing the number of snapshot *T*.

### 5.4. Discussion

The simulation results presented in Section 5.2 and Section 5.3 show that our method attains more satisfactory gain-phase error and DOA estimation performance with respect to the eigenstructure method [13], diagonal line method [17] and MUSIC nulling method [25]. The simulation was carried out in a MATLAB 2020a environment using an Intel 3.1-GHz processor with 8 GB of RAM and under a Windows 10 operating system. All the simulation results were obtained by averaging over 500 independent Monte Carlo trials. The eigenstructure method requires a two-step iteration, the first one involving the gain-phase error estimation and the second, calculating the DOA estimates. As such, a convergence to a global minimum and a unique solution may not been guaranteed, which typically leads to the performance degradation of the parameter estimation. The diagonal line method makes use of the different diagonal lines of the data covariance matrix to estimate the gain-phase errors. Nevertheless, it is known that the diagonal elements of the covariance matrix are mostly contaminated by the environment noises. As shown in Figure 8, in contrast to the Diagonal line method, the proposed method is not sensitive to the environment noises. It obtains a satisfactory estimation performance of the gain-phase error, even when the SNR is 0 dB, due to one high-power calibration source. The MUSIC nulling method requires multiple calibration sources and needs to solve a set of linear equations. Due to the adjustable CSR and the optimal spatial location of the calibration source, our proposed method can offer relatively better estimates than the MUSIC nulling method, which does not provide results about the adjustable CSR and optimal spatial location. However, the proposed method requires a high CSR (larger than 20 dB), and the estimation of covariance matrix $Q_{e}$ in (33) leads to the increase in the number of signal snapshots as compared with these three competing methods.

Our future work is to validate the effectiveness and efficiency of the proposed method by using NI software-defined radio radio testbed. The testbed will work with real Wi-Fi data under IEEE 802.11az for indoor localization and positioning. In this prototype, the gain and phase errors are mainly caused by imperfections in antennas, feedlines, and RF chains. Thus, integrating the proposed calibration procedure in the testbed is helpful in the application for the DOA estimation problem.

## 6. Conclusions

Based on the adaptive antenna nulling technique, a new gain-phase error pre-calibration method is proposed, requiring only one calibration signal with known direction. We divide a ULA with *M* array elements into M−1 sub-arrays, and the explicit expression of gain-phase errors can be derived exactly by applying the SAUNSVs of each sub-array. In order to estimate the SAUNSVs, we develop a WTLS algorithm, and the solution to the WTLS problem in the statistical sense is also studied. These analyses suggest that (1) the optimal location of a unique calibration source is the normal direction of the ULA, and (2) the proposed one requires the calibration signal power to be much larger than the source signal. Simulation results demonstrate the good agreement between the theoretical analyses and experimental predictions, the efficiencies and superiority of the proposed method in terms of the estimation precision of gain-phase errors and DOAs.

## Figures and Tables

**Figure 1 sensors-23-02544-f001:**
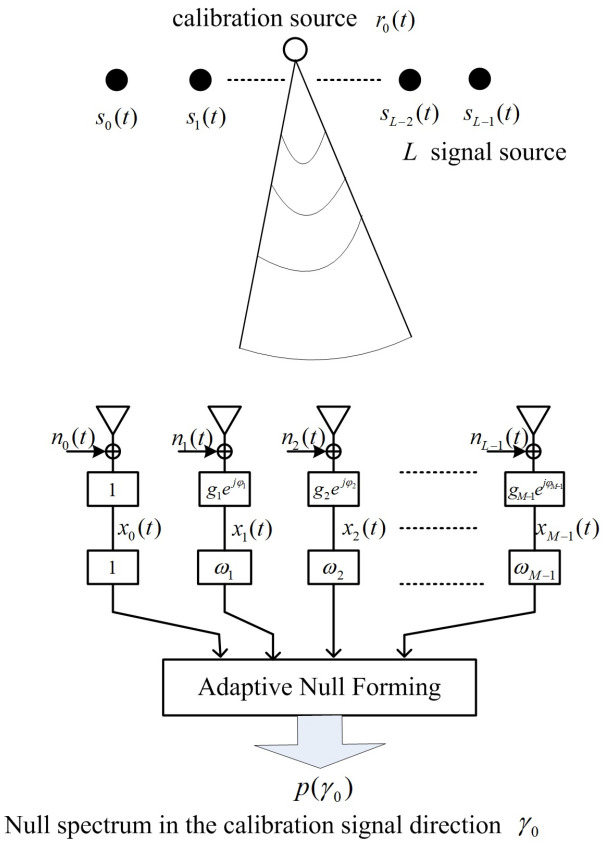
Adaptive antenna nulling array for gain-phase error estimation using one calibration source.

**Figure 2 sensors-23-02544-f002:**
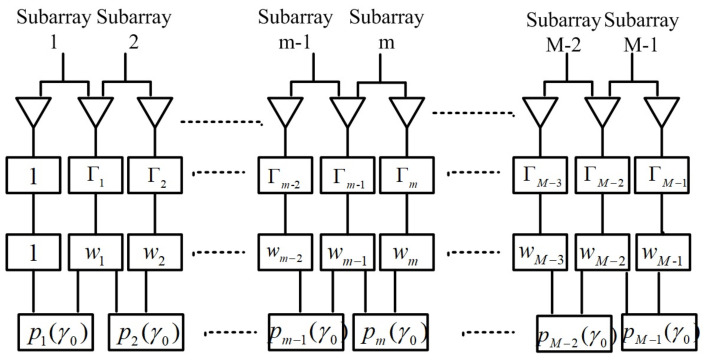
The diagram of the proposed method.

**Figure 3 sensors-23-02544-f003:**
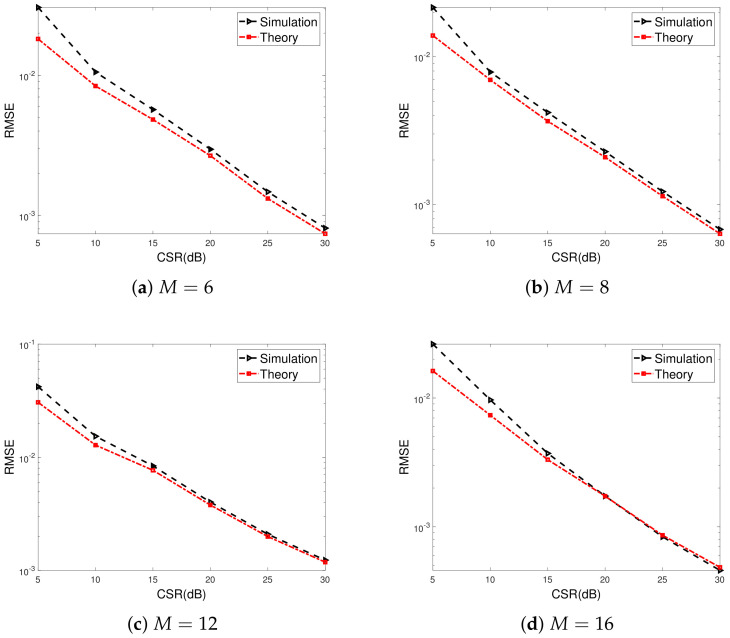
Gain-phase error estimation performances of the proposed method for small-scale ULAs versus CSRs, SNR=40dB, $K_{0}=20$, T=80, N=40. ($K_{0}$: the number of random error vectors, *N*: the number of signal snapshots needed to form one random error vector without the calibration signals, *T*: the number of signal snapshots needed to estimate the SAUNSV with the calibration signals, *M*: the number of array elements).

**Figure 4 sensors-23-02544-f004:**
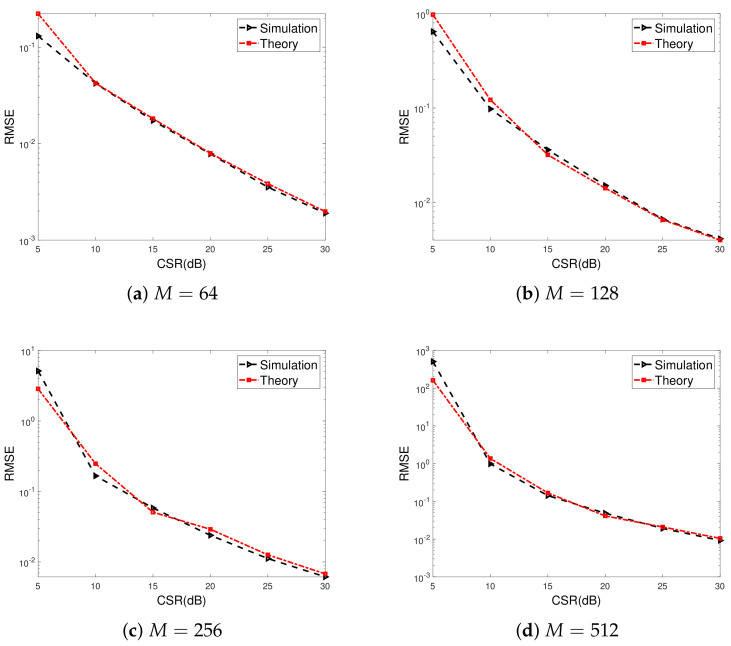
Gain-phase error estimation performance results of the proposed for large-scale ULAs versus CSRs, SNR=40dB, $K_{0}=20$, T=80, N=40.

**Figure 5 sensors-23-02544-f005:**
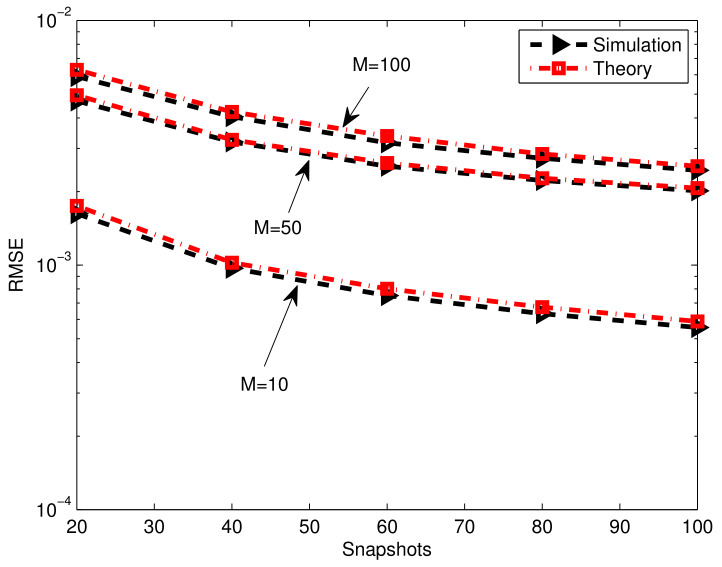
Gain-phase error estimation performance results of the proposed method versus snapshots, CSR=30dB, SNR=40dB, $K_{0}=20$.

**Figure 6 sensors-23-02544-f006:**
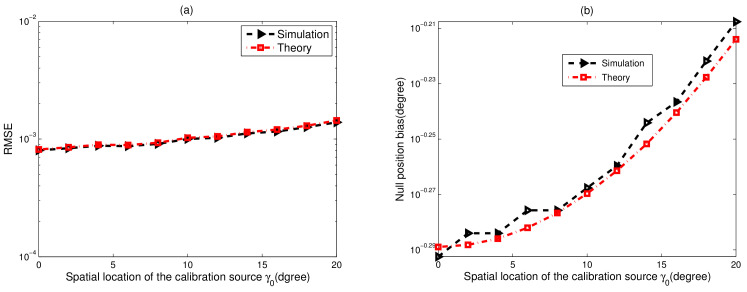
For different spatial locations of the calibration source $\gamma_{0}$, CSR=30dB, SNR=40dB, $K_{0}=20$, T=80, N=40. (**a**) Gain-phase error estimation performance results of the proposed method, M=6. (**b**) Null position bias of the null-spectrum $\Delta \gamma_{0}$, M=2.

**Figure 7 sensors-23-02544-f007:**
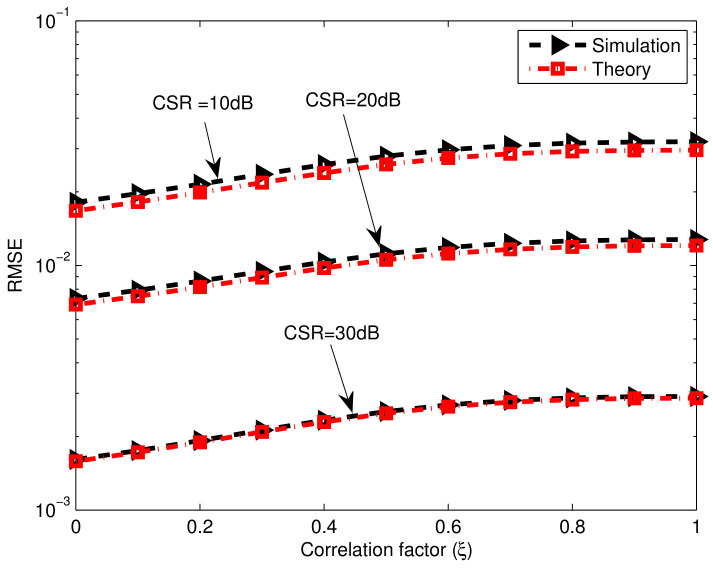
Gain-phase error estimation performances of the proposed method versus correlation factors of source signals, SNR=40dB, M=16, $K_{0}=20$, T=80, N=40.

**Figure 8 sensors-23-02544-f008:**
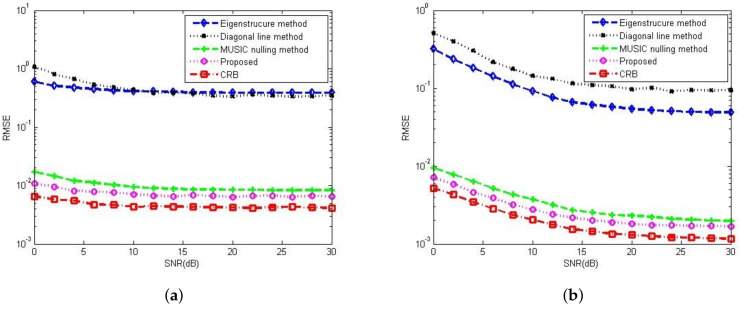
RMSE performance comparison among the Eigenstructure method [13], Diagonal line method [17] and MUSIC nulling method [25] versus SNR, CSR=30dB, $K_{0}=20$, T=80, N=40. (**a**) In 5-element ULA with deterministic gain-phase errors. (**b**) In 16-element ULA with random gain-phase errors.

**Figure 9 sensors-23-02544-f009:**
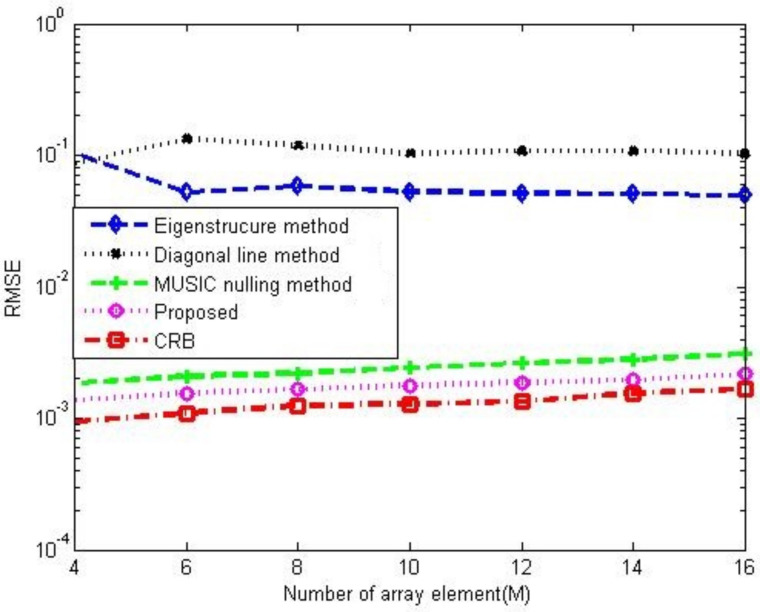
RMSE performance comparison among the corresponding gain-phase estimation methods versus number of array element in the case of random gain-phase errors, CSR=30dB, SNR=20dB, $K_{0}=20$, T=80, N=40.

**Figure 10 sensors-23-02544-f010:**
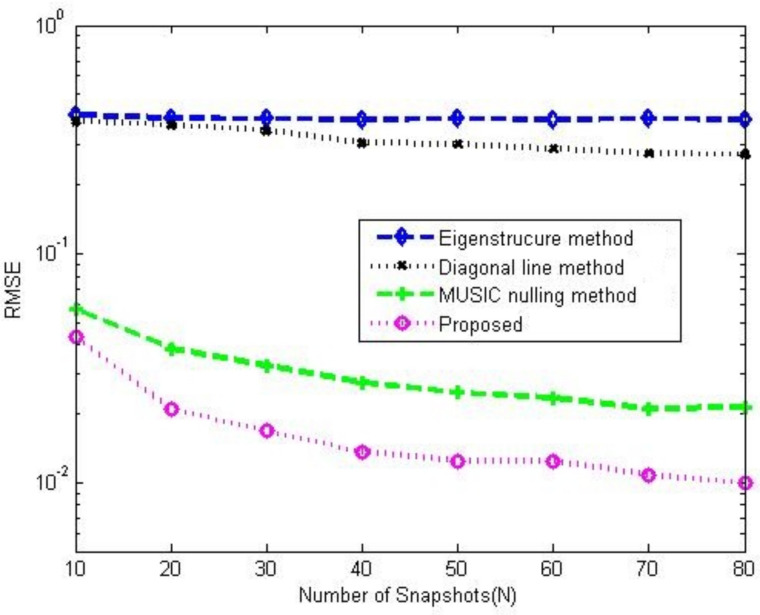
RMSE performance comparison among the corresponding gain-phase estimation methods versus number of snapshot in 5-element ULA with deterministic gain-phase errors, CSR=30dB, SNR=20dB, $K_{0}=20$.

**Figure 11 sensors-23-02544-f011:**
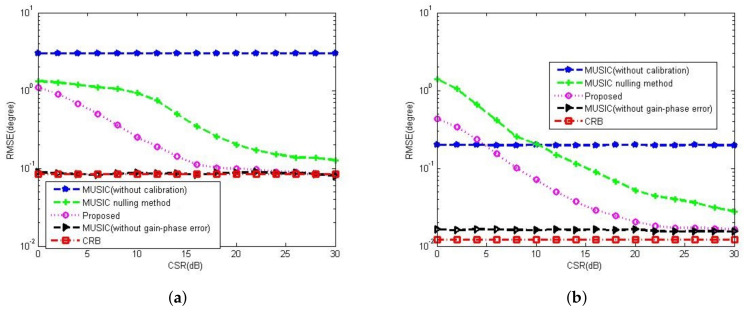
DOA estimation performance with different CSRs, SNR=20dB, $K_{0}=20$, T=80, N=40, $T_{0}=40$. (**a**) In 5-element ULA with deterministic gain-phase errors. (**b**) In 16-element ULA with random gain-phase errors.

**Figure 12 sensors-23-02544-f012:**
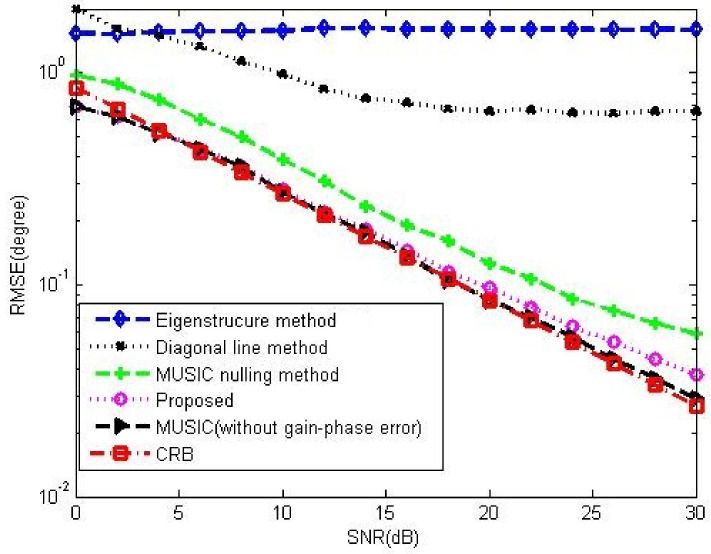
DOA performance comparison among the corresponding methods versus SNR in 5-element ULA with deterministic gain-phase errors, CSR=30dB, $K_{0}=20$, T=80, N=40, $T_{0}=40$.

**Figure 13 sensors-23-02544-f013:**
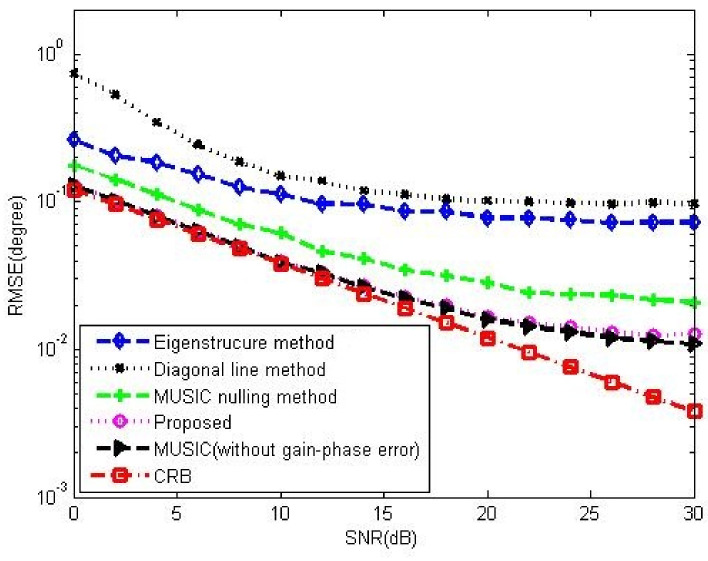
DOA performance comparison among the corresponding methods versus SNR in 16-element ULA with random gain-phase errors, CSR=30dB, $K_{0}=20$, T=80, N=40, $T_{0}=40$.

**Figure 14 sensors-23-02544-f014:**
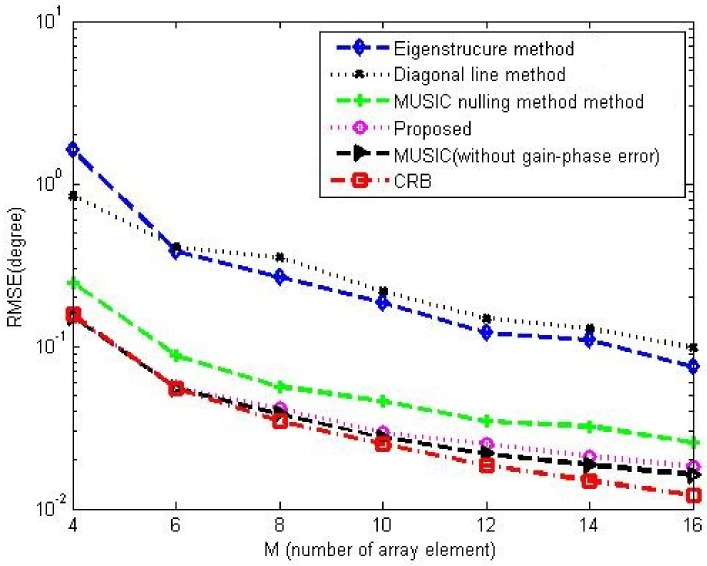
DOA performance comparison among the corresponding gain-phase estimation methods versus number of array element in the case of random gain-phase errors, CSR=30dB, SNR=20dB, $K_{0}=20$, T=80, N=40, $T_{0}=40$.

**Figure 15 sensors-23-02544-f015:**
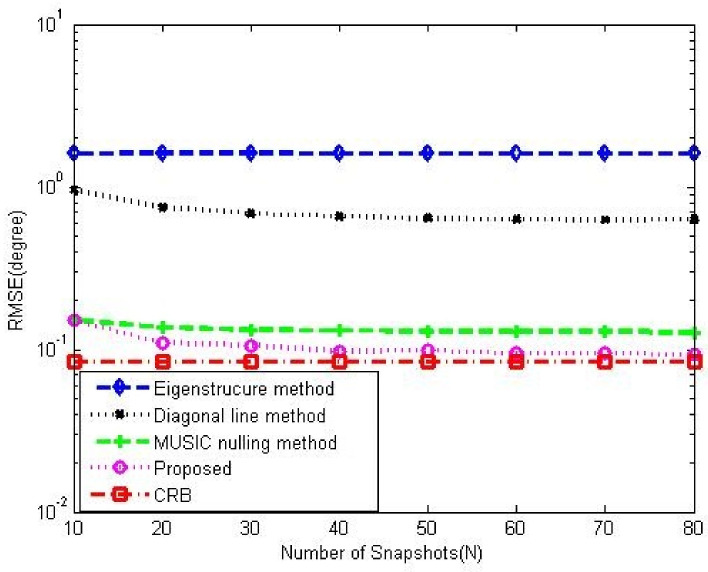
DOA performance comparison among the corresponding gain-phase estimation methods versus number of snapshot in 5-element ULA with deterministic gain-phase errors, CSR=30dB, SNR=20dB, $K_{0}=20$, $T_{0}=40$.

## Data Availability

Not applicable.

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
