# Peer review of "A New Gain-Phase Error Pre-Calibration Method for Uniform Linear Arrays"

_sensors, 2023, doi:10.3390/s23052544_

Round 1

Reviewer 1 Report

In this paper, the authors propose a gain-phase error pre-calibration method for uniform linear arrays (ULAs) based on the adaptive antenna nulling technique, requiring only one calibration source with a known direction of arrival (DOA).

The topic is considered relevant and within the Sensors Journal’s scope. The text is relatively well written, with few spelling errors; the methodology is adequately detailed, the results are consistent, and there are a few considerations for improving the article:

1) The authors must format the paper according to the Sensors template: For example, the first section starts with “0. Introduction,” and it should be “1. Introduction”;

2) The text must be revised for spelling errors:

Ex.: Line 17: “hotspots”; Line 458 “estimation”; Line 464 “… known directions …”; Line 467: “…limits the estimation…”; Line 487: “…method and propose method …”; Line 488: “…can be obtained by:”;

3) Lines 126 and 127. Instead of “3) Some statistical analyses …” it would be better to use: “3. Some statistical and comparative analyses…” because the authors use RMSE, SNR;

4) Line 134: The correct is “Superscripts” and not “Subscripts”;

5) Equation 1: The diag(.) notation was not included in Line 134;

6) Some variables were not defined in the text: Ex.: sigma^2 in Line 156; w0 = 1 in the proposed algorithm 1, “w” in Equation 32, etc.

7) It is suggested to increase the font sizes of the texts used in all Figures. It is not possible to read the text in some of them!

8) It is suggested to use Figure 1 to clarify subsections 1.1 and 1.2, showing some variables (Ex.: internal range, L, c(t), s(t), n(t), r(t), etc.;

Figure 3 can clarify the text of section 3 if the authors define the variables used there; Ex. The variables used in Equation 9;

9) Use a period at the end of Equation 26 instead of a comma;

10) Lines 397, 399, 411, 456, 480: Subsections A, B, and C are not in the text;

11) Line 541: Some abbreviations were not included: LEA, SBAC, CSA, CRB, GESPR, QR-RLS, UMA

12) The authors need to discuss their results with other works adequately. They claim the superiority of their work to some state-of-the-art gain-phase error calibration approaches in the abstract. Thus, they must improve their discussion to prove it.

Considering that the work presents consistent results and the adopted methodology is relatively innovative, it is recommended to accept this work for publication in the journal Sensors after the indicated corrections.

Author Response

Response to the Reviewers’ Comments

Reviewer #1

(1). The authors must format the paper according to the Sensors template: For example, the first section starts with “0. Introduction,” and it should be “1. Introduction”;

Response:

Thank you very much for the valuable comment.

It has been revised.

(2). The text must be revised for spelling errors:

Ex.: Line 17: “hotspots”; Line 458 “estimation”; Line 464 “… known directions …”; Line 467: “…limits the estimation…”; Line 487: “…method and propose method …”; Line 488: “…can be obtained by:”;

Response:

Thank you very much for the valuable comment.

All the spelling errors have been revised.

(3).  Lines 126 and 127. Instead of “3) Some statistical analyses …” it would be better to use: “3. Some statistical and comparative analyses…” because the authors use RMSE, SNR;

Response:

Thank you very much for the valuable comment.

Revised.

(4).  Line 134: The correct is “Superscripts” and not “Subscripts”;

Response:

Thank you very much for the valuable comment.

Revised.

(5). Equation 1: The diag(.) notation was not included in Line 134;

Response:

Thank you very much for the valuable comment.

The diag(.) has been defined in Notation part in Section 1.

(6).  Some variables were not defined in the text: Ex.: sigma^2 in Line 156; w0 = 1 in the proposed algorithm 1, “w” in Equation 32, etc.

Response:

Thank you very much for the valuable comment.

The have been defined in Line 212.

(7).  It is suggested to increase the font sizes of the texts used in all Figures. It is not possible to read the text in some of them!

Response:

Thank you very much for the valuable comment.

The font sizes have been increased.

(8).  It is suggested to use Figure 1 to clarify subsections 1.1 and 1.2, showing some variables (Ex.: internal range, L, c(t), s(t), n(t), r(t), etc.;

Response:

Thank you very much for the valuable comment.

In Figure 1, some variables have been clarified.

(9).  Use a period at the end of Equation 26 instead of a comma;

Response:

Thank you very much for the valuable comment.

Revised.

(10).  Use a period at the end of Equation 26 instead of a comma;

Response:

Thank you very much for the valuable comment.

Revised.

(11).  Line 541: Some abbreviations were not included: LEA, SBAC, CSA, CRB, GESPR, QR-RLS, UMA

Response:

Thank you very much for the valuable comment.

All the abbreviations have been added

(12). The authors need to discuss their results with other works adequately. They claim the superiority of their work to some state-of-the-art gain-phase error calibration approaches in the abstract. Thus, they must improve their discussion to prove it.;

Response:

Thank you very much for the valuable comment.

In the revised version, we discuss our results with the corresponding works in Subsection 5.4.

We would like to acknowledge the anonymous reviewers for their valuable comments and suggestions, which have helped us to improve the paper significantly.

Reviewer 2 Report

Attached Separately.

Author Response

Response to the Reviewers’ Comments

Reviewer #2

(1). One of the imperfections in the sonar array is maintaining the accurate l/2 separations between sensors while mounting them on the array. However, there is no indication that the authors have considered this term in their computation.

Response:

Thank you very much for the valuable comment.

Under assumption of accurate l/2 separation, we derive the proposed method. If the location errors are considered, there are more phase rotation in Equ.(1). The location errors can be included in the phase errors.

(2). What are the other possible array imperfections? Those may be listed. How many of them are considered here for phase error computation?

Response:

Thank you very much for the valuable comment.

All the possible array imperfections are listed in Line 29 and all the origins of gain-phase errors are provided.  

(3).  There are many chances that the random acoustic fluctuations in the medium like a highly dynamic ocean can introduce phase error in the output of the ULA. Authors may comment on it concerning their work.

Response:

Thank you very much for the valuable comment.

This point is commented in Line 65.

(4).  What are the chances that the experimental implementation may not tally with the simulation results reported by the authors? Authors may discuss it in the revised manuscript

Response:

Thank you very much for the valuable comment.

The experimental platform is setting up. However, in the early promotion day, the progress is slow, even once stagnant because of lack of fund.

Our future work is to validate the effectiveness and efficiency of the proposed method by using NI software-defined radio radio testbed. The testbed will work with real Wi-Fi data under IEEE 802.11az for indoor localization and positioning. In this prototype, the gain and phase errors are mainly caused by imperfections in antennas, feedlines, and RF chains. Thus, integrating the proposed calibration procedure in the testbed is helpful in the application for DOA estimation problem.

(5).  The repeatability of the process and the results are important to be discussed in the manuscript with supporting discussions.

Response:

Thank you very much for the valuable comment.

The simulation has been carried out in MATLAB 2020a environment using an Intel 3.1-GHz processor with 8 GB of RAM and under Windows 10 operating system. All the simulation results are obtained by averaging over 500 independent Monte Carlo trials. Thus, from the perspective of numerical simulation, the results can be completely repeated.

We would like to acknowledge the anonymous reviewers for their valuable comments and suggestions, which have helped us to improve the paper significantly.

Reviewer 3 Report

In this paper, Based on the adaptive antenna nulling technique, a new gain-phase error pre-calibration method is proposed, requiring only one calibration source with known direction of arrival (DOA):

This is a very well written paper with state of the art method.

However, only theoritical results are shown in the paper.The quality of the paper can be further improved if the results can be validated by designing antenna hardware.

Author Response

Response to the Reviewers’ Comments

Reviewer #3

In this paper, Based on the adaptive antenna nulling technique, a new gain-phase error pre-calibration method is proposed, requiring only one calibration source with known direction of arrival (DOA):

This is a very well written paper with state of the art method.

However, only theoritical results are shown in the paper. The quality of the paper can be further improved if the results can be validated by designing antenna hardware.

Response:

Thank you very much for the valuable comment.

The experimental platform is setting up. However, in the early promotion day, the progress is slow, even once stagnant because of lack of fund.

Our future work is to validate the effectiveness and efficiency of the proposed method by using NI software-defined radio radio testbed. The testbed will work with real Wi-Fi data under IEEE 802.11az for indoor localization and positioning. In this prototype, the gain and phase errors are mainly caused by imperfections in antennas, feedlines, and RF chains. Thus, integrating the proposed calibration procedure in the testbed is helpful in the application for DOA estimation problem.

We would like to acknowledge the anonymous reviewers for their valuable comments and suggestions, which have helped us to improve the paper significantly.